# FOXJ1 mediates taxane resistance through regulation of microtubule dynamics

Fang Xie [1,4], Ada Gjyrezi [2,4], Daniel Fein[1], Maryam Labaf[1], Larysa Poluben[1], Betul Ersoy-Fazlioglu [1], Christopher M. Dennehy[1], Olga Voznesensky[1], Aniket Gad[1], Eva Corey [3], Andreas Varkaris[1], David J. Einstein[1], Rupal S. Bhatt [1], Paraskevi Giannakakou [2,5] ✉ & Steven P. Balk [1,5] ✉

Docetaxel is the first-line chemotherapy for metastatic prostate cancer (PC), but clinically meaningful mechanisms of resistance remain to be established. Here we show, in an in vivo model of docetaxel resistant PC patient-derived xenografts, increased expression of genes that drive development of multi-ciliated cells including *FOXJ1* and its effectors, many of which regulate microtubules (MTs). Mechanistically, FOXJ1 overexpression confers docetaxel resistance in vitro and in vivo, which is associated with decreased docetaxel-mediated MT bundling. Overexpression of a MT-associated FOXJ1-regulated gene (*TPPP3*) has similar effects. Conversely, FOXJ1 knockdown impairs basal MT function, enhances taxane binding to MTs, and increases docetaxel sensitivity. These results establish mechanistic causality between the FOXJ1 signaling axis, MT biology, and taxane resistance. Clinically, *FOXJ1* gene amplification is increased in taxane-treated PC patients. Moreover, in the CHAARTED clinical trial of docetaxel combined with androgen deprivation for metastatic PC, higher baseline FOXJ1 is predictive of decreased survival in PC patients treated with docetaxel, further supporting clinical relevance. Together, these findings identify a previously unrecognized clinically impactful mechanism of taxane resistance whose exploitation could stratify patients who will not benefit from taxane treatment.

Taxanes used in the clinic (docetaxel, paclitaxel, and cabazitaxel), either as single agents or in combination therapies, have efficacy in multiple solid tumors. In prostate cancer (PC), taxanes (docetaxel or cabazitaxel) are the only class of cancer chemotherapeutics that prolong survival as single agents in men with metastatic castration-resistant PC (mCRPC), defined as patients who have progressed after standard androgen deprivation therapy (ADT, surgical or medical castration)[1,2]. Moreover, a subset of men who become resistant to docetaxel may subsequently respond to the related taxane, cabazitaxel, consistent with the clinical efficacy of taxanes in this disease. In

addition, in men with metastatic castration-sensitive PC, the combination of docetaxel with androgen receptor (AR) targeted therapy has shown unprecedented clinical efficacy, causing a paradigm shift in clinical practice[3–6]. Thus, docetaxel is not only used as standard-of-care treatment in men with mCRPC but is also increasingly used earlier, in combination with ADT, or with ADT plus androgen receptor (AR) pathway inhibitors (triple therapy), as the initial treatment in men with castration-sensitive PC. However, while docetaxel clearly enhances responses to ADT, the extent to which it enhances responses to ADT plus an androgen receptor (AR) pathway inhibitor in all or subsets of

[1]Department of Medicine, Division of Oncology and Cancer Center, Beth Israel Deaconess Medical Center, Boston, MA, USA. [2]Department of Medicine, Division of Hematology & Medical Oncology, Weill Cornell Medical Center, New York, NY, USA. [3]Department of Urology, University of Washington School of Medicine, Seattle, Washington, USA. [4]These authors contributed equally: Fang Xie, Ada Gjyrezi. [5]These authors jointly supervised this work; Steven P. Balk, Paraskevi Giannakakou. ✉e-mail: pag2015@med.cornell.edu; sbalk@bidmc.harvard.edu

men remains unclear. Moreover, despite this central role of taxanes in PC and other cancers, there is a major gap in our understanding of clinically relevant mechanisms of taxane resistance[7,8].

Taxanes act by binding to the β-subunit of the tubulin αβ heterodimer, stabilizing MTs and suppressing their dynamic properties, which are integral to their function during interphase and mitosis. Thus, taxanes cause aberrant mitotic arrest in cycling cells while also disrupting myriad other MT-dependent functions in interphase, such as translocation of nuclear proteins, including the AR in PC cells[9,10]. Mechanisms underlying taxane resistance, which inevitably emerge across all clinical settings, have been extensively studied in vitro. However, none of the multiple reported mechanisms of resistance have been shown to be clinically relevant. These include overexpression of drug efflux transporters such as *ABCB1*[11–15], tubulin mutations at the drug's binding site, or overexpression of the *TUBB3* tubulin isoform. They also include alterations in pathways downstream of MT stabilization that allow cells to survive prolonged mitotic arrest or disruption of MT dynamics during interphase, such as increased expression of PLK1, BCL2, or MCL1, lack of functional p53, upregulation of Notch, NFκB activation, or increases in a GATA2-IGF2 network[16–19]. In PC, this may also include expression of ERG or of MT-independent AR splice variants[20–22].

The absence of clinically faithful in vivo models of taxane resistance that recapitulate patients' initial response followed by relapse has contributed to our inability to identify clinically relevant mechanisms of taxane resistance. Herein, we developed an in vivo docetaxel-resistant model using castration-resistant PC patient-derived xenografts (PDXs). Molecular characterization of the resistant tumors identified upregulation of FOXJ1 as a driver of taxane resistance, associated with decreased docetaxel-mediated MT bundling and mitotic arrest. FOXJ1 overexpression caused resistance to docetaxel in vitro and in vivo, while FOXJ1 knockdown impaired basal MT dynamics and sensitized to docetaxel, establishing causality. FOXJ1 has been well-established as a master transcription factor regulating the development of multiciliated cells, where it controls expression of multiple cilia-associated genes[23–25], but previous studies have not implicated FOXJ1 in the regulation of MT dynamics in epithelial cells or in the mechanism of action of taxanes. The findings in this study support a broader role for FOXJ1 in non-ciliated cells, specifically in taxane resistance in PC and beyond.

## Results

### FOXJ1 expression is increased in docetaxel-resistant tumors
Docetaxel is the first-line chemotherapy used for PC treatment, but the mechanisms underlying clinical treatment resistance that inevitably occurs are poorly elucidated. To decipher in vivo molecular pathways mediating docetaxel resistance, we used two PDX models of castration-resistant PC (LuCaP35CR and LuCaP70CR). The PDXs were established in castrated male scid mice and then treated with cycles of docetaxel, using a clinically used schedule for PC (30mg/kg, i.p., every 21 days). Both the LuCaP35CR and LuCaP70CR PDXs responded initially to docetaxel but relapsed after several cycles of treatment (Fig. 1A, B). We harvested resistant tumors 2 weeks after their last dose of docetaxel, along with control tumors of approximately the same volume that were never treated. We then performed RNA-seq and low-pass whole-genome sequencing (LP-WGS) on docetaxel-resistant and -sensitive tumors.

Analysis of differentially expressed genes in the LuCaP70CR PDXs showed a dramatic increase in mRNA encoding the drug exporter protein ABCB1/MDR1 (Fig. S1A). Previous studies have shown that *ABCB1* gene amplification or overexpression can confer resistance to multiple large lipophilic drugs, including taxanes, which provided validation for our approach. However, the importance of this mechanism in patients remains to be established, and examination of genomic data from tumor biopsies from taxane-treated PC patients

did not show an increased frequency of ABCB1 amplification as compared to tumors from patients who were not taxane treated (Fig. S1B). Interestingly, while increased expression of ABCB1 is generally mediated by gene amplification, by LP-WGS we found that the *ABCB1* gene was already amplified in the initial LuCAP70CR PDX prior to docetaxel treatment (Fig. S1C). This amplification was presumably driven by prior therapy in the patient and indicates that the increased gene expression in the docetaxel-resistant PDXs was due to an epigenetic mechanism.

Analysis of differentially expressed genes in the second docetaxel-resistant PDX, LuCaP35CR, did not show increased *ABCB1* (Fig. 1C). Instead, the most significantly increased gene was *GMNC* (geminin coiled-coil domain-containing), which encodes the transcription factor GEMC1 that initiates the development of multiciliated cells[24–26]. Along these lines, FOXJ1, a downstream target of GEMC1 and master regulator of ciliogenesis that controls expression of multiple MT-associated proteins[23,25], was also significantly enriched. Consistent with these findings, examination of Gene Ontology Biological Process (GOPB) gene sets showed enrichment for gene sets directly related to the MT cytoskeleton, including Microtubule Bundle Formation and Motile Cilium Assembly (Fig. 1D, E). Amongst the enriched genes common to these gene sets was *FOXJ1*, as well as multiple genes encoding cilia-related proteins that are positively regulated by FOXJ1 in multiciliated cells (*TPPP3, DNAAF3, SPEF1, ZMYND10, SPAG17*, and *SPAG1*)[23,25].

Taken together, these results suggested that activation of a ciliogenesis program through the GEMC1 and FOXJ1 axis in the docetaxel-resistant LuCaP35CR xenografts contributed to docetaxel resistance. However, the functions of FOXJ1 have been most extensively studied in multiciliated cells and in cells that develop mono-motile cilia, and have not been well-studied in nonciliated cells[23–25,27,28]. To address this, we next assessed for correlations between FOXJ1 and these cilia-related genes by mining transcriptomic data from patients with localized PC in TCGA. We found significant correlations between FOXJ1, its upstream regulator GEMC1, and multiple of the FOXJ1 target genes that we also found increased in the docetaxel-resistant LuCaP35CR PDXs, such as *ZMYND10, SPEF1, TPPP3, DNAAF3, SPAG17*, and *SPAG1* (Fig. 1F, Figs. S2, S3A). GEMC1 expression was also correlated with a subset of cilia-related genes, but the correlations were greater with FOXJ1, consistent with FOXJ1 directly regulating these genes (Fig. S3B, C). Together, these data support FOXJ1 regulation of MT-associated proteins in non-ciliated cells, including in PC cells.

To further assess FOXJ1 as a mediator of taxane resistance across tumor types we examined DepMap drug response data on approximately 300 cancer cell lines, which showed that higher FOXJ1 expression was correlated with docetaxel resistance across multiple tumor types (Fig. 1G, Fig. S4A). To assess the potential relevance of FOXJ1 and downstream genes in docetaxel resistance in PC patients, we examined the SU2C dataset of patients with advanced PC whose metastatic castration-resistant tumors were biopsied and analyzed by whole-exome sequencing[29]. Notably, tumors from patients who had been treated with a taxane (primarily docetaxel) had a significantly higher frequency of *FOXJ1* gene amplification (10.88%) compared with tumors from patients who were not exposed to a taxane (5.45%) (Fig. 1H). FOXJ1 mRNA levels were also increased in the taxane-exposed patients, although the increase was not significant, which may reflect biopsies being obtained at variable times after treatment was discontinued (Fig. S4B).

### FOXJ1 overexpression decreases docetaxel-induced MT bundling and confers docetaxel resistance in vitro
To further assess whether FOXJ1 plays a functional role in docetaxel resistance, we generated FOXJ1 stably over expressing LNCaP cells (Fig. 2A). Colony formation assays showed that FOXJ1 overexpressing (OE) LNCaP cells were less sensitive to docetaxel treatment than LNCaP cells containing empty vector control (EV) (Fig. 2B, C, S5A). A major effect of taxanes in vitro is impairment of mitotic spindle function,

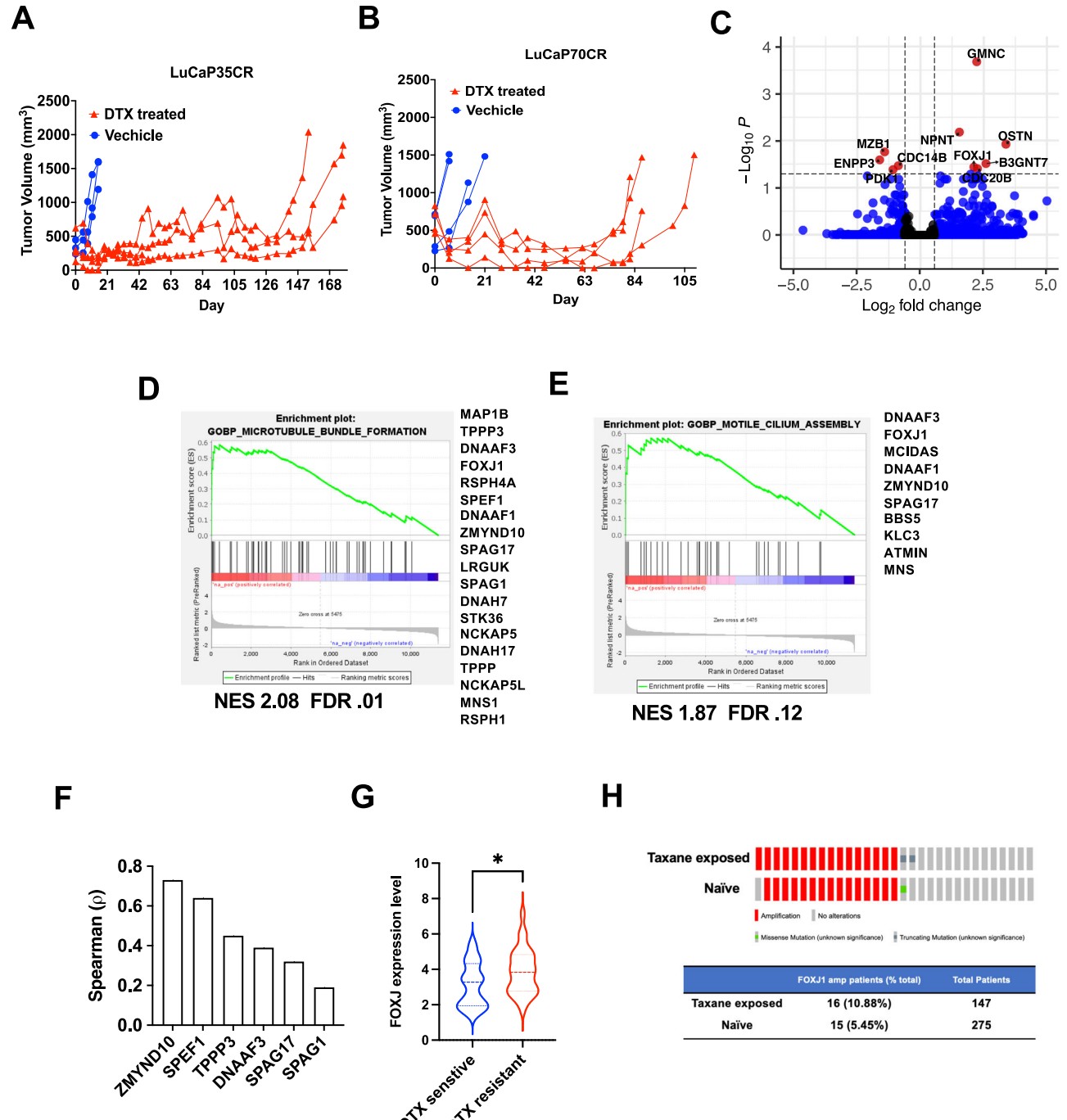

**Fig. 1 | RNA-seq analysis of resistance pathways in docetaxel-resistant tumors.**
**A, B** Mice with LuCaP35CR (A) or LuCaP70CR (B) PDXs were treated with either vehicle or docetaxel (DTX) at 30 mg/kg by i.p. injection every 21 days starting at day 0 as indicated. Docetaxel and vehicle-treated xenografts were harvested when they reached ~1500 mm³. The treated tumors were harvested 2 weeks after their final dose of docetaxel. **C** Differential gene expression for LuCaP35CR in harvested resistant xenografts versus vehicle-treated xenografts. The most significantly altered genes are labeled. **D, E** GSEA applied to LuCaP35CR-resistant versus vehicle-treated xenograft tumors for microtubule bundle formation and motile cilium assembly pathways. The significantly increased genes in each gene set are listed in rank order. **F** Spearman correlation for the co-expression of FOXJ1 with the indicated genes in the TCGA PC dataset. **G** Cancer cells expressing FOXJ1 ( > 1.5 fold above average) were assessed for DTX sensitivity above (sensitive) and below (resistant) the median sensitivity in the DepMap dataset. Error bars indicate the standard error of the mean (SEM), * $p < 0.05$ ($p = 0.0321$, Mann-Whitney test, two-tailed). **H** FOXJ1 amplification in the SU2C patients that were previously taxane-treated versus naïve. Source data are provided as a Source Data file.

with subsequent mitotic arrest and apoptosis. Flow cytometry further showed that there was less accumulation of cells in G2/M in FOXJ1_OE than in EV control cells after docetaxel treatment, indicating that FOXJ1 overexpression could attenuate taxane-induced mitotic arrest (Fig. 2D, E). To further determine effects on mitosis, we examined

phosphorylation of serine 10 on histone 3 (pH3S10) as an indicator of mitotic cells. Treatment with 10 nM docetaxel for 6 h caused a marked increase in pH3S10 in the control cells, but not in the FOXJ1 over-expressing cells, where 100 nM docetaxel was required to obtain comparable levels of pH3S10 (Fig. 2F).

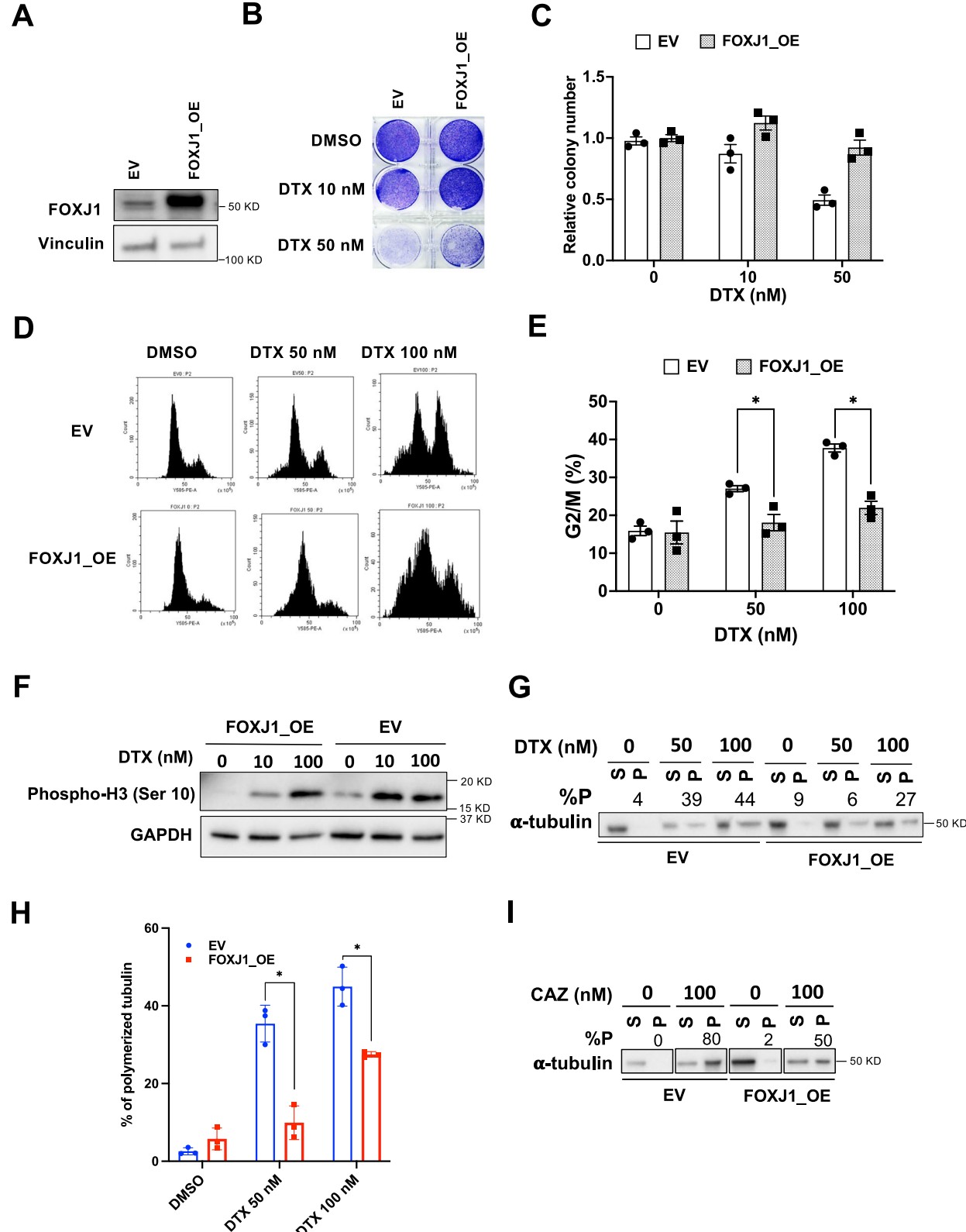

Taxanes act by promoting tubulin polymerization leading to the formation of stable MT bundles in interphase cells. Based on our previous finding that docetaxel-resistant cells had decreased docetaxel-induced MT bundling[30], we hypothesized that FOXJ1 over-expression might be acting by impairing docetaxel-driven MT poly-merization. To test this hypothesis, we examined effects on soluble versus polymerized tubulin after docetaxel treatment. Interestingly,

the fraction of polymerized tubulin (% P) under basal conditions was modestly increased in the FOXJ1 overexpressing cells (Fig. 2G, H). As expected, polymerized tubulin was greatly increased after docetaxel treatment in the control LNCaP EV cells. In contrast, docetaxel-driven tubulin polymerization was significantly impaired in FOXJ1-overexpressing cells. We also examined another taxane, cabazitaxel, which has efficacy in a subset of PC that becomes resistant to

**Fig. 2 | FOXJ1 overexpression confers resistance to docetaxel in vitro. A** LNCaP cells with stable expression of FOXJ1 (FOXJ1_OE) or empty vector control (EV) were immunoblotted for FOXJ1 or vinculin. The blot is representative of three independent experiments. **B** Image of violet purple stained colonies for FOXJ1_OE and EV cells treated with DMSO and indicated concentrations of DTX. The image is representative of three independent experiments. **C** Colonies for FOXJ1_OE and EV LNCaP cells in the indicated treatment conditions were counted and normalized based on vehicle treated EV controls. Mean and standard deviation in triplicate wells from one of three representative experiments are shown. **D** Representative cell cycle profiles for FOXJ1_OE and EV cells treated for 24 h with vehicle (DMSO) or indicated concentrations of DTX. **E** Percentage of cells in G2/M phase for FOXJ1_OE and EV cells treated with DMSO or indicated concentrations of DTX for 24 h, mean +/− SD, $n = 3$ independent experiments, * $p < 0.05$ ($p = 0.026$ and $p = 0.0075$,

Student's t-test, two-tailed) compared to EV. **F** FOXJ1_OE and EV cells were treated with docetaxel for 24 h and immunoblotted for H3S10 phosphorylation. The blot is representative of three independent experiments. **G** Tubulin polymerization assay showing amount of α-tubulin in pellet fraction (P) and soluble fraction (S) in representative experiment. The P% was calculated based on grayscale intensity of western blot bands for each treatment group [P% = P/( P+ S)*100%]. The blot is representative of three independent experiments. **H** P% values from three biological repeats for the indicated treatment group in FOXJ1_OE and EV cell, mean +/− SD, * $p < 0.05$ ($p = 0.019$ and $p = 0.034$, Student's t-test, two-tailed), compared to EV. **I** Western blot demonstrated the amount of α-tubulin from pellet fraction (P) and soluble fraction (S) in FOXJ1_OE versus control cells after cabazitaxel (CAZ) treatment. The blot is representative of three independent experiments. Source data are provided as a Source Data file.

docetaxel. Notably, FOXJ1 overexpression also reduced MT aggregation in response to cabazitaxel (Fig. 2I). Together these results demonstrate that FOXJ1 overexpression confers docetaxel resistance by impairing the ability of the drug to effectively engage with and alter its target.

We also assessed the effects of FOXJ1 overexpression on the response to other agents that target tubulin by binding to distinct sites. FOXJ1 overexpression conferred resistance to each of these agents (colchicine, vincristine, epothilone A, and maytansine) (Fig. S6A). Notably, FOXJ1 overexpressing cells also had decreased sensitivity to olaparib, cisplatin, and carboplatin, although the effect was less marked than for the tubulin-targeted agents (Fig. S6B). The basis for this effect is not clear, but could be due to increased MT dynamics (see below) and subsequent increased drug export, increased response to DNA damage, or other mechanisms.

As noted above, FOXJ1 has been most studied as a driver of multiciliated cells. Therefore, we also assessed for cilia in FOXJ1 overexpressing cells. As a positive control, cilia could be readily detected by immunofluorescence of acetylated tubulin and centrin, a component of the centriole, in BEAS-2B lung bronchial epithelium cells (Fig. S7). In contrast, we did not find cilia or evidence of centriole amplification in LNCaP cells overexpressing FOXJ1.

### FOXJ1 downregulation sensitizes to docetaxel in vitro

We then examined whether downregulation of endogenous FOXJ1 expression affects responses to docetaxel. We used two independent shRNAs to suppress FOXJ1 expression in LNCaP cells (Fig. 3A). Colony formation assays showed that the FOXJ1 knockdown (KD) LNCaP cells were more sensitive to docetaxel treatment than the control cells (Fig. 3B, Fig. S5C). We next examined FOXJ1 expression in a series of PC cell lines, and found the highest expression in DU145 cells (Fig. S8A). Interestingly, we also found that DU145 xenografts were relatively resistant to docetaxel (Fig. S8B). Knockdown of FOXJ1 in DU145 cells (Fig. 3C) increased their sensitivity to docetaxel in colony formation assays (Fig. 3D, Fig. S8C). Consistent with this result, the G2/M fraction was greater in the FOXJ1 KD cells than in the control cells after 24-h docetaxel treatment (Fig. 3E, F). Confocal microscopy of the MT network in the FOXJ1 KD versus the control shNeg cells indicated an increase in docetaxel-induced MT bundling in the FOXJ1 KD cells (Fig. 3G). To quantify this response to docetaxel, we used an algorithm that was developed and clinically validated to assess effective drug-target engagement (DTE) between taxanes and MTs, and subsequent increases in MT bundling, by measuring the microtubule fluorescence intensity (MFI) in mitotic and interphase MTs[30]. This confirmed increased MT bundling in the docetaxel-treated FOXJ1 KD cells (Fig. 3H). Together, these results show that the downregulation of FOXJ1 expression sensitizes cells to docetaxel.

One basis for this sensitization could be increased taxane binding in the FOXJ1 KD cells. To test this hypothesis, we used Flutax, a fluorescent taxane derivative, to directly assess taxane

binding on interphase MTs. Permeabilized cells were treated for 5 minutes with Flutax, which was then washed away and Flutax-MT binding was followed by time-lapse live cell imaging (Fig. S9A). Flutax binding was greater at time 0 and in all subsequent times in the FOXJ1 KD cells, while the rate of Flutax dissociation from MTs was similar (Fig. 3I, J, Fig. S9B). The pattern of the MT cytoskeleton decorated by bound Flutax was also strikingly distinct, with the FOXJ1 KD cells displaying a more extensive labeling of the MT network. These findings indicate that FOXJ1 depletion increases the availability of taxane binding sites, likely increasing the drug's association rate, but does not have a clear effect on the drug's dissociation rate.

### FOXJ1 regulates basal MT dynamics

Tubulin acetylation at lysine 40 (K40ac) in α-tubulin is the sole post-translational modification that marks the MT lumen, accumulates on long-lived stable MTs, and alters the conformation of MTs to enhance their flexibility and prevent structural damage[31,32]. A hallmark of taxane activity is an increase in tubulin acetylation, consistent with taxane-mediated suppression of MT dynamics and an increase in MT stability. Notably, we found that acetylated-α-tubulin (K40ac) was decreased upon FOXJ1 depletion, indicating that FOXJ1 and its transcriptional network may regulate MT polymerization dynamics and stability under basal conditions (Fig. 4A). As expected, docetaxel treatment caused a dose-dependent increase in acetylated α-tubulin across all conditions. However, docetaxel caused a dramatic fold increase in acetylated- α-tubulin in the FOXJ1 KD cells, up to 8.7 fold higher than their untreated counterparts, compared to only an 1.8-fold increase in cells expressing endogenous FOXJ1. These results are consistent with the increased taxane binding and sensitivity in the FOXJ1 KD cells, but also point to a previously unrecognized role of FOXJ1 in basal MT dynamics in non-ciliated cells.

To further assess the potential effects of FOXJ1 on MT dynamics, we transfected the FOXJ1 KD or control cells with fluorescently labeled EB1, a protein that binds to the plus ends of growing MTs. This generates distinct "comet tails" that represent the visible length of time an EB1 molecule remains bound to a growing MT plus end, and is used as a read-out of MT dynamics. Live-cell imaging of untreated versus docetaxel-treated cells, and tracking of the number of EB1 comets bound at the tips of individual growing MTs, did not reveal any changes caused by FOXJ1 depletion under basal conditions (Fig. 4B, C, Movie S1). As expected, taxane treatment decreased the number of EB1 comets in the control cells, consistent with taxane's ability to suppress MT dynamics resulting in EB1 dissociation from MTs. Notably, this taxane-mediated decrease was more pronounced in the FOXJ1 KD cells. This result indicates that the taxane is targeting a greater fraction of growing MTs in these cells, and is consistent with the Flutax binding studies above.

While the baseline number of EB1 comets was not detectably altered by FOXJ1 depletion, there was a decrease in EB1 comet length,

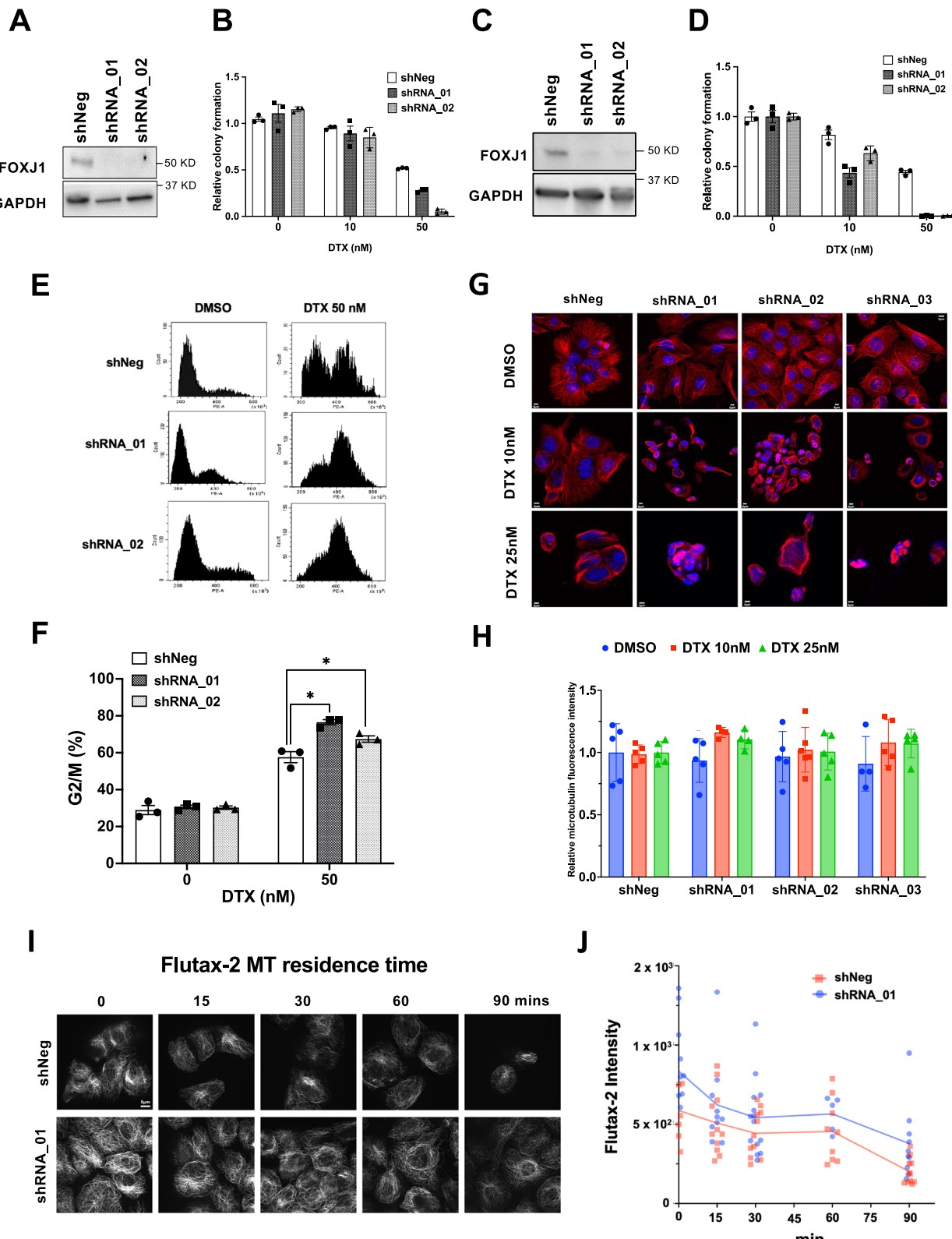

suggesting an effect on MT growth rates, as the comet length is directly related to the speed at which the MT is growing (Fig. 4D, Fig. S10)[33]. Next, we assessed MT growth rates, using live cell time-lapse microscopy to calculate EB1 comet speed. The FOXJ1 KD cells had a decrease in EB1 comet speed relative to control cells under basal conditions and in response to taxane (Fig. 4E, Movie S2). In agreement with the decrease in MT growth rates in FOXJ1 KD cells, we also observed a concomitant decrease in the MT path/trajectory length calculated by quantifying the length of time that EB1 remained bound at MT plus ends, marking growing MTs (Fig. 4F, G, Movie S3). Taxane treatment markedly decreased the MT trajectory length, with a more pronounced effect in FOXJ1 KD cells, consistent with their enhanced taxane sensitivity. Together, these results indicate that FOXJ1 depletion modulates basal MT dynamics by decreasing MT growth rates while also augmenting MT stabilization in response to taxane treatment.

**Fig. 3 | FOXJ1 downregulation sensitizes to docetaxel in vitro. A** LNCaP cells with stable expression of lentivirus encoding independent FOXJ1 shRNA (shRNA_01 and shRNA_02) or non-targeting control shRNA (shNeg) were immunoblotted for FOXJ1. The blot is representative of three independent experiments. **B** Colonies for LNCaP cells with FOXJ1 knockdown and shNeg in the indicated treatment conditions were counted and normalized based on shNeg vehicle (DMSO) control. Mean and standard deviation in triplicate wells from one of three representative experiments are shown. **C** Stable FOXJ1 knockdown in DU145 (shRNA_01 and shRNA_02) and non-targeting negative control in DU145 (shNeg) cells were immunoblotted for FOXJ1. The blot is representative of three independent experiments. **D** Colonies for DU145 cells with FOXJ1 knockdown and shNeg in the indicated treatment groups were counted and normalized based on shNeg DMSO control. Mean and standard deviation in triplicate wells from one of three representative experiments are shown. **E** Cell cycle profiles for DU145 cells with FOXJ1 knockdown or shNeg control treated with vehicle (DMSO) or DTX at 50 nM for 24 h. The profiles are representative of three independent experiments. **F** Percent of G2/M phase cells for DU145 cells expressing FOXJ1 or control shRNA treated with DMSO or DTX at 50 nM for 24 h, mean +/− SD, * $p < 0.05$, compared to shNeg by

paired Students t-test in three independent experiments (shNeg vs shRNA-01, $p = 0.031$; shNeg vs shRNA-02, $p = 0.043$) **G** Immunofluorescence images for tubulin (red) and DAPI stained nuclei (blue) in DU145 with FOXJ1 knockdown or shNeg in the indicated treatments for 24 h. The images are representative of three independent experiments. **H** MT fluorescence intensity (MFI) was quantified in DU145 shNeg control cells and FOXJ1-knockdown DU145 cells (shRNA_01, shRNA_02, shRNA_03) using ImageJ. For each treatment condition, five microscopy fields were analyzed, and MFI values were normalized to cell number. Data are mean +/− SD of technical replicates (fields, $n = 5$) from a representative experiment of three independent experiments. **I** Native cytoskeletons from DU145 FOXJ1 or control shRNA cells were treated with FITC-conjugated paclitaxel (Flutax-2 at 1 μM) for 5 minutes. Flutax-2 was then removed, and residence time on microtubules was quantified following wash-out by time-lapse imaging. The images are representative of three independent experiments. **J** Quantification of Flutax-2 fluorescence intensity at each time point in multiple fields (5 −12 fields examined under each condition at each time point from the experiment in Fig. 4I as indicated in the figure), followed by linear regression analysis. Source data are provided as a Source Data file.

Conversely, these data indicate that increased FOXJ1 may enhance basal MT dynamics in interphase cells. To test this, we examined the androgen-stimulated nuclear import of AR, which, similar to other nuclear proteins, is MT-dependent. As expected, AR nuclear localization in LNCaP cells was increased by treatment with a potent agonist (R1881) (Fig. S11). Notably, this nuclear localization was increased in LNCaP cells overexpressing FOXJ1, consistent with enhanced MT dynamics. However, the taxane-mediated decrease in AR nuclear localization was not mitigated in the FOXJ1 overexpressing cells. One basis for this persistent suppression of AR nuclear localization by taxane may be dependence on a distinct subset of MTs and MT-associated proteins that have greater sensitivity to taxanes, but further studies are needed to assess this or other mechanisms.

### FOXJ1 overexpression confers resistance to docetaxel in vivo

To determine if increased FOXJ1 can confer docetaxel resistance in vivo, we used the FOXJ1 OE versus EV control LNCaP cells to establish subcutaneous xenografts. When tumors reached approximately 500 mm³, mice were treated with 30 mg/kg docetaxel. FOXJ1 over-expression did not alter growth prior to docetaxel treatment, but markedly impaired the response to the initial dose of docetaxel on day 9, and the second dose on day 30 (Fig. 5A). Immunoblotting of tumors confirmed FOXJ1 overexpression in the OE versus EV control tumors (Fig. 5B). To assess for mechanisms of resistance, additional FOXJ1 OE and EV control xenografts were established and harvested 3 days after a single treatment with docetaxel or vehicle control. Examination of representative sections stained for tubulin indicated that the number of abnormal mitotic figures in response to docetaxel was decreased in the FOXJ1 OE tumors (Fig. 5C).

We then again measured MFI to quantify effective drug-target engagement (DTE) between taxanes and MTs in mitotic and interphase MTs[30]. Interestingly, MFI under basal conditions was higher in the FOXJ1 overexpressing tumors (Fig. 5D), consistent with the higher basal tubulin acetylation levels in control versus FOXJ1 depleted cells (Fig. 4A). In contrast, while docetaxel significantly increased DTE (higher MFI) in the control tumors, the increase in the FOXJ1 over-expressing cells was less marked and did not reach statistical significance. We also analyzed tumors from OE and EV mice sacrificed at ~3 days and ~10 days after the second dose of docetaxel. This analysis similarly showed increased DTE in the EV control tumors at 3 days versus 10 days after treatment, but not in the FOXJ1 overexpressing tumors (Fig. 5E). These findings further indicate that while FOXJ1 overexpression may increase MT stability under basal conditions, it mitigates taxane-induced MT stabilization.

We further examined another group of FOXJ1_OE versus EV xenografts by RNA-seq prior to therapy to identify pathways that may

be increased by FOXJ1. The most significantly enriched Hallmark gene set was Mitotic Spindle (FDR 16%) (Fig. S12A), suggesting that a subset of FOXJ1-regulated genes may contribute to spindle formation. Analysis of the Gene Ontology Biological Process gene sets showed enrichment for one cilium-related gene set (Regulation of Cilium Assembly, FDR 59%), but no highly enriched gene sets (FDR < 25%) (Fig. S12B). Notably, a substantial fraction of the genes (229 genes, 29%), that were increased in the docetaxel-resistant LuCaP35CR xenografts (791 genes total) were also increased in the FOXJ1_OE xenografts (Fig. 5F). This overlap supports the conclusion that increased FOXJ1 was modulating the transcriptome and contributing to docetaxel resistance in the LuCaP35CR xenografts. Consistent with the established function of FOXJ1 as a regulator of ciliogenesis, these 229 genes were most associated with positive regulation of cilium assembly (Fig. 5G).

### TPPP3 is regulated by FOXJ1 and its overexpression confers resistance to docetaxel

We hypothesized that taxane resistance is mediated by FOXJ1 through the downstream genes it regulates. We recently reported that expression of an alternatively spliced short isoform of CLIP170 contributed to taxane resistance in gastric cancer[34], but did not find expression of this isoform in cells overexpressing FOXJ1 (Fig. S13). TPPP3 (tubulin polymerization promoting protein 3) is a microtubule-associated protein and downstream target of FOXJ1 in multiciliated cells, and was increased in the docetaxel-resistant LuCaP35CR PDXs. Moreover, we confirmed that TPPP3 protein was increased by FOXJ1 in the FOXJ1_OE LNCaP cells (Fig. 6A). To further assess the potential role of TPPP3 in docetaxel resistance, we generated TPPP3 overexpressing LNCaP cells (Fig. 6B) and performed in vitro functional studies. Flow cytometry showed that TPPP3 overexpression mitigated the docetaxel-mediated increase in G2/M fraction (Fig. 6C). Tubulin polymerization assays showed decreased accumulation of polymerized tubulin (%P) in TPPP3_OE LNCaP after docetaxel treatment compared to EV control (Fig. 6D, E). These in vitro results indicated that TPPP3, similar to FOXJ1, could impair docetaxel-driven MT polymerization.

To determine whether increased TPPP3 could confer docetaxel resistance in vivo, mice bearing TPPP3_OE, FOXJ1_OE, or EV control xenografts were treated with 30 mg/kg docetaxel on day 0 and day 14. Consistent with the results in Fig. 3A (carried out in an independent cohort), overexpression of FOXJ1 impaired the response to docetaxel (Fig. 6F). The overexpression of TPPP3 similarly diminished the response to docetaxel, although to a lesser degree, suggesting that other FOXJ1 downstream targets also contribute to drug resistance. To assess the basis for resistance, we established additional LNCaP_TPPP3 and EV xenografts and harvested them on day 3 after an initial

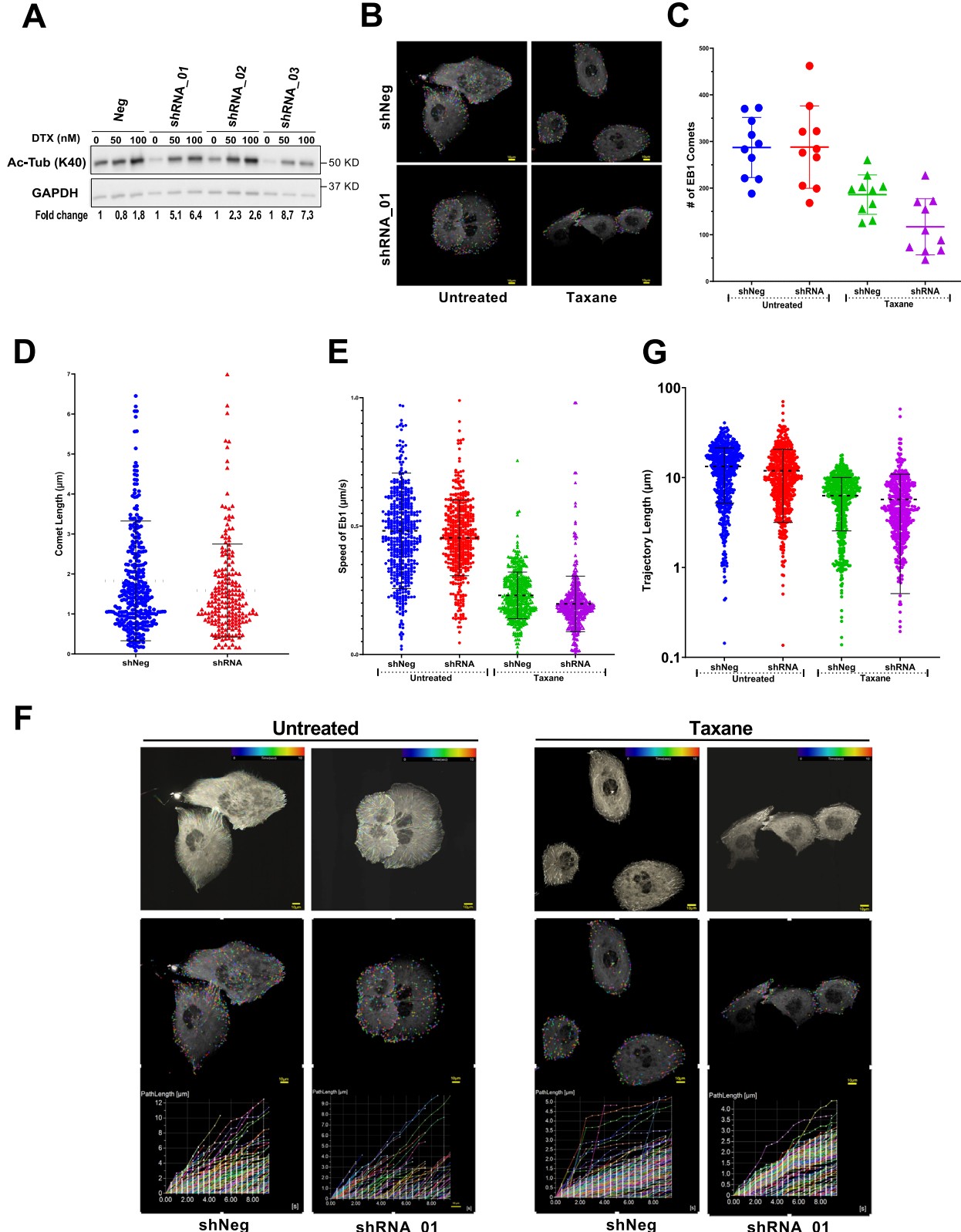

docetaxel treatment, and did fluorescence imaging for tubulin (Fig. 6G). This indicated a decrease in docetaxel-driven MT bundling and multipolar mitotic spindle formation in the LNCaP_TPPP3 versus EV control cells, which was confirmed by measurement of MFI (Fig. 6H). Together these findings indicate that FOXJ1 mediates doc-etaxel resistance, at least in part, through increased TPPP3.

**FOXJ1 expression increases early in response to docetaxel**
We initially examined the LP-WGS data to determine the basis for increased FOXJ1 expression in the docetaxel-resistant LNCaP35CR PDXs, which did not show *FOXJ1* gene amplification (Fig. S14A). To determine whether taxane-mediated disruption of MT dynamics may feedback to directly alter FOXJ1 expression, we examined FOXJ1

**Fig. 4 | FOXJ1 depletion impairs MT dynamic. A** Immunoblot analysis of acetylated α-tubulin (K40) in DU145 cells expressing control shRNA (Neg) or independent FOXJ1 shRNAs (shRNA_01, shRNA_02, shRNA_03) treated with docetaxel (0, 50, or 100 nM) for 24 h. Band intensities were quantified by densitometry, normalized to GAPDH, and expressed as fold change relative to the corresponding negative control and indicated numerically below each band. The blot is representative of three independent experiments. **B** Grayscale images of EB1-EGFP−expressing DU145 shNeg or FOXJ1-knockdown (shRNA_01) cells, imaged live under basal conditions or following taxane treatment. EB1 comets marking growing MT plus ends are pseudo-colored to aid visualization and tracking; acquisition and analysis parameters (exposure, frame rate, denoising/background subtraction, and detection thresholds) were held constant within each experiment across conditions. The images are representative of three independent experiments. **C** EB1 comet number per cell in control versus treated cells was determined in an experiment that is representative of three biological repeats. Ten cells per condition were quantified; each dot (untreated) or triangle (taxane-treated) represents one cell. The center line denotes the mean and error bars indicate SD. **D** EB1 comet length in control versus treated cells was determined in an experiment that is representative of three biological repeats. Comet length was quantified from 10 cells per condition; each data point represents an individual comet ($n \approx 250$), with dashed lines indicating mean ± SD. Mean comet length was 1.83 μm in shNeg cells and 1.59 μm in shRNA_01 cells. **E** EB1 comet growth speed (μm/sec) was quantified from time-lapse imaging of EB1 comet length in control versus treated cells in an experiment that is representative of three biological repeats. Each point represents an individual comet ($n = 450–500$ comets per condition), shown to visualize distribution breadth; horizontal bars denote mean ± SD. Mean speeds were 0.48 μm/s (untreated shNeg) vs 0.45 μm/s (untreated shRNA_01), decreasing to 0.23 μm/s (taxane-treated shNeg) and 0.20 μm/s (taxane-treated shRNA_01). **F** EB1 comet trajectory was mapped in control versus treated cells in an experiment that is representative of three biological repeats. Panels (left to right) show untreated shNeg, untreated shRNA_01, taxane-treated shNeg, and taxane-treated shRNA_01 cells. Top: maximum-intensity projections of EB1 trajectories generated from 10-s time-lapse sequences. Trajectories were generated from time-lapse imaging acquired over a 10-second interval and overlaid to visualize microtubule plus end growth dynamics. Tracks are color-coded using a continuous color spectrum to represent time progression, with blue indicating the initial position of each EB1 comet and red indicating the final position (see color timelapse bar for each panel). Bottom: representative frames with detected comets pseudo-colored to indicate individual tracks; corresponding trajectory traces or path lengths (μm) are shown below. The x-axis represents time (s) and the y-axis displays the path length (μm) for each EB1 trajectory. Maximum trajectory lengths reached 12 μm (untreated shNeg) and 9.5 μm (untreated shRNA_01), decreasing to 5 μm and 4.5 μm after taxane treatment, respectively. **G** Quantitative representation of the results shown in F. EB1 trajectory length (μm) was quantified from time-lapse imaging (450–500 comets per condition). Mean trajectory lengths were 13.30 μm (untreated shNeg) vs 11.96 μm (untreated shRNA_01), decreasing to 6.29 μm (taxane-treated shNeg) and 5.70 μm (taxane-treated shRNA_01). Source data are provided as a Source Data file.

expression in response to docetaxel in vitro. Notably, treatment with 10 nM docetaxel caused a time dependent increase in FOXJ1 mRNA after ~24 h that continued for up to 96 h, and treatment with various doses of docetaxel for 96 h caused a dose dependent increase in FOXJ1 mRNA in LNCaP cells (Fig. S14B, C). Similar results were also observed in CWR22Rv1 cells (Fig. S14D). Docetaxel treatment similarly increased FOXJ1 protein in LNCaP and CWR22Rv1 cells cultured under standard 2D conditions (Fig. S14E) or under 3D conditions (Fig. S14F). Consistent with induction of FOXJ1, docetaxel also increased the expression of TPPP3 in LNCaP and CWR22Rv1 cells (Fig. S14G, H).

The time course for FOXJ1 induction suggested that its expression may be cell cycle regulated and that the increased FOXJ1 may be indirect and due to docetaxel-mediated mitotic arrest. To test this hypothesis, we initially assessed whether FOXJ1 was increased in subconfluent *versus* confluent cultures. Significantly, FOXJ1 (and TPPP3) was higher in confluent cultures, indicating it is not increased in cycling cells (Fig. S15A). We next used FACS to isolate untreated DU145 cells in G0/G1 and G2/M. Notably, FOXJ1 mRNA and protein were expressed at comparable levels in G0/G1 versus G2/M cells (Fig. S15B, C). We similarly analyzed LNCaP cells and found comparable levels of FOXJ1 mRNA in G0/G1 and G2/M fractions (Fig. S15D). These observations indicated that the docetaxel-induced increase in FOXJ1 expression was not due to the accumulation of cells in G2/M.

Notably, FOXJ1 is induced by GEMC1 (encoded by *GMNC*) in multiciliated cells, and GEMC1 was increased in the docetaxel-resistant LuCaP35CR cells. Moreover, FOXJ1 expression in PC clinical samples is correlated with GEMC1 (see Fig. S3). Therefore, we next assessed GEMC1 expression in response to docetaxel and found it was induced within 12 h by docetaxel, indicating that its expression is responsive to disruption of microtubule dynamics and is consistent with GEMC1 driving the subsequent expression of FOXJ1 (Fig. S16A). GEMC1 expression in multiciliated cells is initiated by a decrease in Notch signaling. Notably, we found in the TCGA PC gene data set that GEMC1 expression was highly negatively correlated with expression of the Notch-induced genes HES1 and HEY1, suggesting that docetaxel may be inducing GEMC1 by decreasing Notch signaling (Fig. S16B). However, we did not find a decrease in HES1 or HEY1 expression in response to docetaxel, indicating that the mechanism is not decreased Notch (Fig. S16C). Overall, these results support a feedback mechanism through which cells respond to docetaxel by increasing GEMC1 and downstream FOXJ1. However, while this may contribute to intrinsic

resistance in some tumors, acquired resistance is presumably dependent on further adaptations in the expression of a subset of genes downstream of FOXJ1.

## Higher FOXJ1 in patient tumors is predictive of docetaxel resistance

We next addressed whether there was an association between *FOXJ1* expression and tumor behavior. In the TCGA cohort of untreated primary PC, there was no difference in survival in men with tumors expressing FOXJ1 above versus below the median (Fig. S17A). There was also no significant difference when the analysis was limited to higher-grade tumors (Gleason 8 or above) (Fig. S17B), although there was a trend toward decreased survival in the upper versus lower quartile of FOXJ1 expression ($p = 0.098$) (Fig. S17C). Conversely, another primary PC cohort showed a trend toward increased survival in the tumors with above the median FOXJ1 expression (Fig. S18A), and in tumors in the upper versus lower quartile of FOXJ1 expression (Fig. S18B) ($p = 0.094$ and 0.30, respectively) (33). We also analyzed the SU2C cohort of metastatic castration resistant PC (32), which showed a trend towards decreased survival in tumors with above the median (Fig. S19A) or upper quartile of FOXJ1 expression (Fig. S19B) ($p = 0.14$ and 0.15, respectively). Consistent with FOXJ1 not being prognostic, its expression was not correlated with genomic alterations in the tumor suppressor genes *RB1*, *TP53*, or *PTEN* (Fig. S20).

To determine whether FOXJ1 expression may be predictive of response to docetaxel we examined data from the CHAARTED clinical study, in which patients with metastatic castration-sensitive PC were randomized to androgen deprivation therapy (ADT) versus ADT combined with six cycles of docetaxel[3]. Whole transcriptome data from pretreatment tumors, in conjunction with treatment and survival data, were available for analysis from a subset of patients[35]. Similar to the overall cohort, this subset of men showed highly significant improvement in progression-free survival (PFS) and overall survival (OS) from the addition of docetaxel, indicating that this subset of patients is representative of the entire patient cohort (Fig. S21A, B). Interestingly, analysis of the overall cohort (independent of treatment) showed that higher FOXJ1 expression (above the median or upper quartile) was weakly associated with improved PFS and OS, although the differences were not significant (Fig. S22A−D). In the ADT alone cohort, FOXJ1 expression above the median had a stronger trend toward improved PFS and OS ($p = 0.18$ and 0.19, respectively)

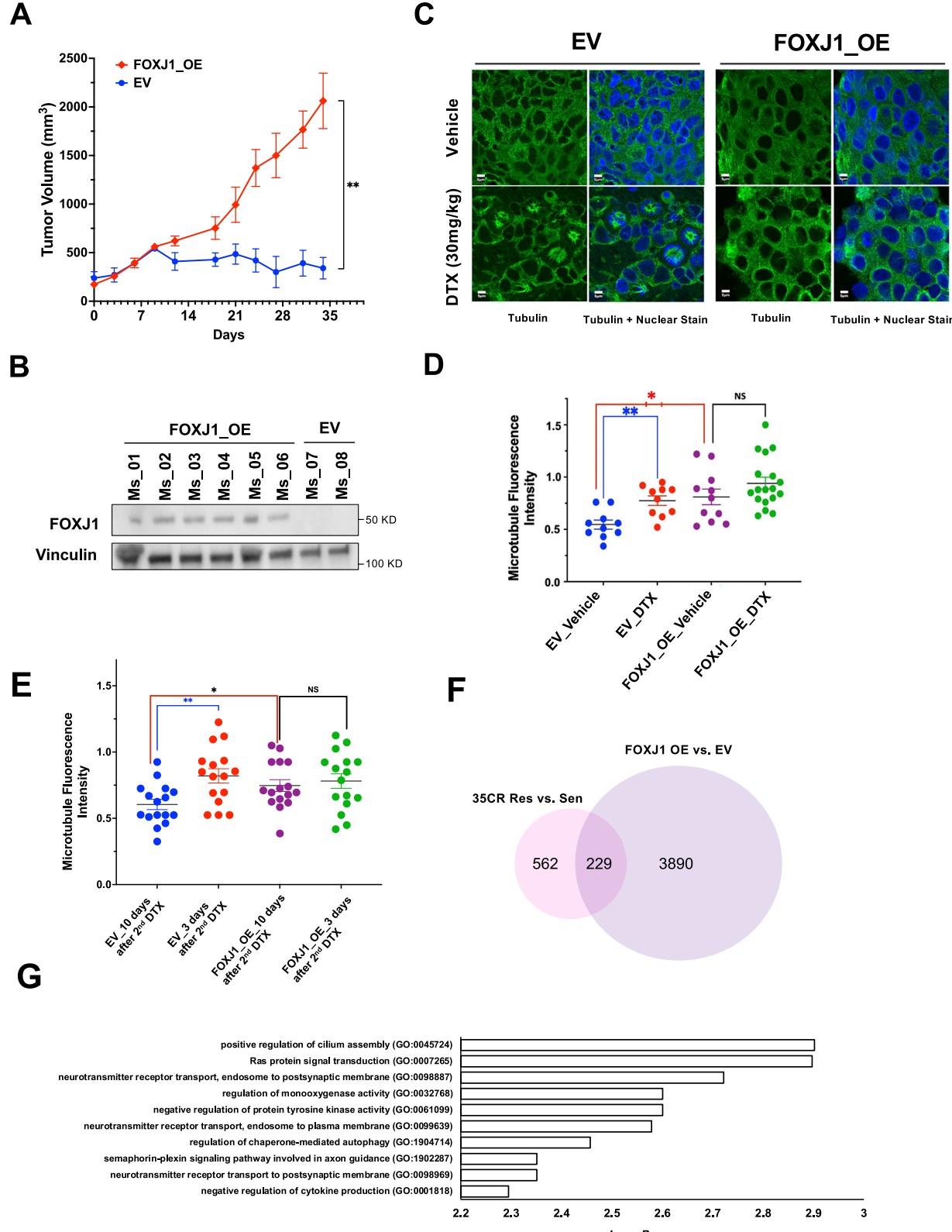

(Fig. S23A, B). In contrast, the trend was reversed in the combination arm, with FOXJ1 expression above the median associated with decreased PFS and OS ($p = 0.17$ and $0.40$) (Fig. S23C, D).

We similarly assessed outcomes in tumors with FOXJ1 expression in the upper versus lower quartiles. Higher FOXJ1 expression was again associated with improved PFS and OS in the ADT alone arm ($p = 0.067$ and $0.024$) (Fig. S24A, B), and with decreased PFS and OS in the combination arm ($p = 0.13$ and $0.13$) (Fig. S24C, D). A final comparison was between tumors with FOXJ1 expression in the upper 25% versus the bottom 75%. Higher FOXJ1 in the ADT alone arm was again associated with improved PFS and OS, which was significant for OS ($p = 0.036$) (Fig. 7A, B). High FOXJ1 in the combination arm of ADT plus docetaxel was significantly associated with decreased PFS and OS ($p = 0.0053$ and $0.052$) (Fig. 7C, D). Notably, comparison of the arms showed

**Fig. 5 | FOXJ1 overexpression confers in vivo resistance to DTX. A** Mice bearing LNCaP FOXJ1_OE and LNCaP EV xenografts were treated with docetaxel (DTX, 30 mg/kg, i.p., 6 mice per each group). The first dose was administered when tumors reached ~500 mm³ (Day 9), and a second dose was of Day 30 (arrows). Tumors were monitored by caliper measurements. **p < 0.01 (p = 0.0037, Two-Way ANOVA, mixed-effects). **B** FOXJ1 expression in 6 individual untreated FOXJ1_OE xenografts and in two untreated EV xenografts with vinculin as loading control. **C** Mice bearing EV and FOXJ1_OE xenografts were treated with one dose of either vehicle or docetaxel (DTX, 30 mg/kg, i.p., 3 mice for each group). After 3 days, mice were sacrificed and tumors were harvested. Representative immunofluorescence images for tubulin (green) and nuclei (blue) are shown. **D** MFI scores for each group in (C) were quantified based on at least 10 visions per each group (visions were from 3 mice in each group) and data are shown as individual field measurements with mean ± SD. Statistical significance was assessed using the two-sided Mann–Whitney test. In LNCaP EV tumors, MFI was significantly higher after treatment with DTX (p = 0.0338). In FOXJ1-overexpressing tumors, MFI did not differ significantly after DTX treatment (p = 0.0587). When untreated, FOXJ1 OE tumors showed

significantly higher MFI compared with EV tumors (p = 0.0482). **E** MFI scores were measured in the xenografts from (A) treated with two doses of DTX that were sacrificed at ~10 days after the second dose or in additional mice sacrificed at ~3 days after the final dose. At least 10 visions were captured for each group (visions were from 3 mice in each group) and data are shown as individual field measurements with mean ± SD. Statistical significance was assessed using the two-sided Mann–Whitney test. In LNCaP EV tumors, MFI was significantly higher at 10 days compared with 3 days after the second DTX dose (p = 0.0019). In FOXJ1-overexpressing tumors, MFI did not differ significantly between the 10-day and 3-day post-treatment groups (p = 0.1625). At the 10-day time point, FOXJ1 OE tumors showed significantly higher MFI compared with EV tumors (p = 0.0058). **F** Venn diagram showing the overlapping differentially expressed genes (DEGs) based on the RNA-seq between FOXJ1_OE versus EV tumors from (A), and LNCaP35CR Resistant versus Sensitive tumors from Fig. 1. There are 229 overlapping genes. **G** Enrichr analysis of the 229 overlapping genes showing the top gene pathways based on the p-value. Source data are provided as a Source Data file.

---

dramatic PFS and OS benefits to addition of docetaxel in tumors with lower FOXJ1 expression, but no evident benefit in the tumors with high FOXJ1 (Fig. 7E, F).

We carried out a similar analysis for correlations between OS in CHAARTED and expression of *RB1*, *PTEN*, and *TP53*, which did not reveal any correlation (Fig. S25). We similarly did not find an association between OS and expression of *ABCB1*. These findings are consistent with our mechanistic studies and strongly support the conclusion that increased FOXJ1 is a clinically significant mediator of taxane resistance, and suggest that high FOXJ1 expression may have value as a biomarker to identify patients who are less likely to benefit from the addition of docetaxel to ADT.

## Discussion

FOXJ1 is a master transcription factor that regulates multiple MT-associated proteins and drives the development of both multiciliated and motile monociliated cells, but its functions in nonciliated cells have not been well studied. In this study, we found that expression of FOXJ1, in conjunction with cilium-related genes regulated by FOXJ1 in multiciliated cells, was increased in docetaxel-resistant LuCaP35CR PDXs. Mechanistically, we then found that FOXJ1 overexpression in vitro could mitigate docetaxel effects on colony formation and mitosis, and in vivo could confer docetaxel resistance. Mechanistically we found that FOXJ1 overexpression decreased levels of docetaxel-induced MT polymerization. Conversely, RNAi-mediated knockdown of FOXJ1 increased docetaxel sensitivity based on colony formation, fraction of cells in G2/M, and MT polymerization, and this was associated with increased taxane binding to MTs.

Notably, FOXJ1 depletion under basal conditions decreased MT growth rates and tubulin acetylation, a MT posttranslational modification that occurs in the MT lumen and is associated with long-lived stable MTs. While there is a debate as to whether tubulin acetylation confers MT stability or is a consequence of MT stabilization, as this modification accumulates on stabilized MT polymers, a recent report using the NCI-60 panel of cancer cell lines showed that basal levels of tubulin acetylation do not correlate with taxane cytotoxicity[36], thereby uncoupling acetyl-tubulin from taxane sensitivity. However, our results showed a decrease in basal acetyl-tubulin upon FOXJ1 depletion, which was correlated with increased taxane binding to MTs and cytotoxicity. A possible mechanistic explanation for these findings is suggested by a recent report showing that the alpha-tubulin acetyl-transferase (ATAT) binding site in the MT lumen partially overlaps the taxane binding site, and that paclitaxel competes with ATAT for MT binding[37]. These findings support a model where the low acetyl-tubulin levels in the FOXJ1-depleted cells reflect lower occupancy of the taxane binding sites in the MT lumen by ATAT, favoring increased taxane binding. While further studies are needed to test this or alternative

mechanisms, the results indicate that FOXJ1 has a physiological role in the regulation of MT dynamics in non-ciliated cells, with implications for taxane sensitivity.

The FOXJ1-mediated mitigation of taxane effects presumably reflects altered expression of MT-associated proteins downstream of FOXJ1. One such protein that is FOXJ1-regulated in multiciliated cells, and was increased in the docetaxel-resistant LuCaP35CR PDXs, is TPPP3. We confirmed that FOXJ1 overexpression increased TPPP3 in PC cells, and found that TPPP3 overexpression (similar to FOXJ1 overexpression) could mitigate the effects of docetaxel on cell cycle and MT aggregation, and decrease sensitivity to docetaxel in vivo. However, FOXJ1-overexpressing xenografts were more resistant, consistent with FOXJ1 overexpression being mediated by altered expression of additional MT-associated genes. Interestingly, overexpression of FOXJ1, but not TPPP3, increased basal MT bundling, also consistent with FOXJ1 regulation of MT dynamics through effects on additional MT-associated proteins.

The CHAARTED and STAMPEDE trials showed a benefit of adding docetaxel to ADT as initial therapy for metastatic castration-sensitive PC[3,4]. The recent ARASENS and PEACE-1 trials have further indicated that patients with metastatic castration-sensitive PC and high disease burden benefit from upfront "triple therapy" with the combination of ADT, androgen synthesis/signaling inhibitors (abiraterone or darolutamide), and docetaxel[5,6]. Although clinically efficacious, taxanes causes significant decline in quality of life during the treatment course and can lead to life-long toxicities[38]. Therefore, there remains an unmet need to identify factors predictive of taxane resistance and response to better personalize therapy in the upfront setting. Using data from the CHAARTED study, we found that higher FOXJ1 expression in pretreatment biopsies was predictive of worse PFS and OS in patients treated with combination ADT and docetaxel, but not in the ADT alone arm. Moreover, men with upper quartile FOX1 expression had comparable PFS and OS in both arms, indicating that these men had no significant benefit from the addition of docetaxel. Overall, these findings support the conclusion that FOXJ1 is a clinically relevant mediator of taxane resistance. Further investigations are warranted to determine whether FOXJ1 expression may be useful for identifying patients who are less likely to benefit from taxanes in combination with ADT or in other contexts, and to identify therapies that can interfere with FOXJ1 or specific downstream effectors, thereby enhancing taxane efficacy.

## Methods
### Study approval

All animal experiments were approved by the BIDMC Institutional Animal Care and Use Committee (IACUC Protocol #051-2022-25) and were performed in accordance with institutional and national

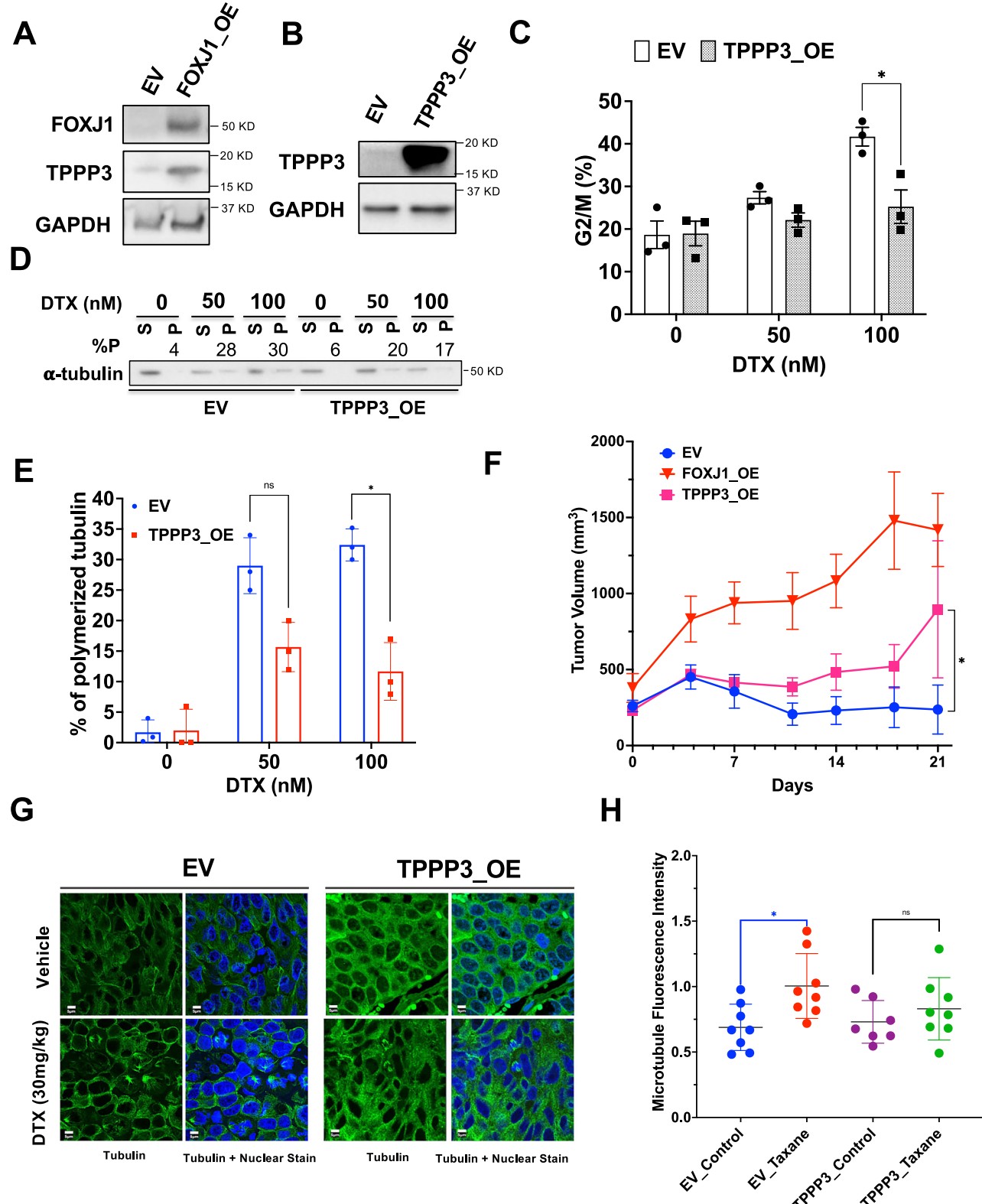

guidelines. All mice used were male as PC only occurs in males. All mice were sacrificed if tumors exceeded 2000 mm³.

## Reagents

Docetaxel (T1034), cabazitaxel (T2543), vincristine (T6721), colchicine (T0320), maytansine (T21351), epothilone (T6490), olaparib (T3015), cisplatin(T1058), and carboplatin (T1564) were purchased from TargetMol for in vitro cell culture use. For the in vivo study, docetaxel was obtained from SelleckChem USA and prepared in saline containing 5% DMSO, 30% PEG300, and 5% Tween 80 at a concentration of 2.0 mg/mL.

## Plasmids and cell lines

FOXJ1 overexpression (FOXJ1_OE), TPPP3 overexpression (TPPP3_OE), and empty vector (EV) plasmids, which were constructed on a lentivirus-based vector (pReceiver-Lv241), were

**Fig. 6 | TPPP3 is regulated by FOXJ1 and its overexpression confers partial in vivo resistance to DTX. A** TPPP3 expression in LNCaP FOXJ1_OE and EV cells. The blot is representative of three independent experiments. **B** LNCaP cells stably expressing TPPP3 (TPPP3_OE) or empty vector control (EV). The blot is representative of three independent experiments. **C** The percentage in G2/M phase for TPPP3_OE and EV cells treated with the indicated concentrations of DTX for 24 h in three independent experiments, $*p = 0.023$, compared to EV. **D** Western blot demonstrating the amount of α-tubulin in the pellet fraction (polymerized, P) and soluble fraction (S) that were separated by tubulin polymerization assay. The % polymerized was calculated based on grayscale intensity of western blot bands for each treatment group [P% = P/($p$ + S)*100%]. The blot is representative of three independent experiments. **E** P% values from three biological replicates for the indicated treatment group for TPPP3_OE and EV, SD, $* p < 0.05$ ($p = 0.0928$ at 50 nM, $p = 0.038$ at 100 nM), compared to EV. **F** Mice bearing FOXJ1_OE, LNCaP TPPP3_OE, or LNCaP_EV xenografts were treated with docetaxel (DTX, 30 mg/kg, i.p., 6 mice per each group, the first dose on Day 0, the second dose on Day 14) (arrows) and monitored until day 21. $*p < 0.05$ ($p = 0.03$, Two-Way ANOVA, mixed-effects). **G** Mice bearing EV and TPPP3 _OE xenograft tumors were treated with one dose of either vehicle or docetaxel (DTX, 30 mg/kg, i.p., 3 mice for each group). After three days, mice were sacrificed, and tumors were harvested. Representative immunofluorescence images from 3 mice for tubulin (green) and nuclei (blue) in tumors treated as indicated are shown. **H** MFI was quantitified by ImageJ for TPPP3_OE and EV in the indicated treatment groups from (**G**), at least 8 visions (visions were from 3 mice in each group) were taken for each group and data are shown as individual field measurements with mean ± SD. In LNCaP EV tumors, MFI was significantly higher after treatment with DTX ($p = 0.0148$). In TPPP3-overexpressing tumors, MFI did not differ significantly after DTX treatment ($p = 0.3357$). Source data are provided as a Source Data file.

purchased from GeneCopoeia, Rockville, MD. The set of FOXJ1 shRNA plasmids, including TCRN0000420413 (shRNA-01), TCRN0000015305 (shRNA-02), and TCRN0000015305 (shRNA-03), with a non-target scramble, was purchased from Sigma-Aldrich. LNCaP, DU145, CWR22Rv1, and 293 T cells were obtained from ATCC (Manassas, VA), authenticated by short tandem repeat profiling, and were free of mycoplasma.

### Generation of cell lines with stable EB1
Approximately 1 × 106 cells of the DU145 shNeg and DU145 FOXJ1 KD (shRNA_01) cell lines were infected with EB1-EGFP lentiviral particles (VectorBuilder Cat. # VB230531-1518prn) at a multiplicity of infection of 1. Clones were picked using cloning cylinders, grown individually, and screened by live microscopy. Specific clones showing EB1-EGFP expression were subsequently used with the Tubulin Tracker Deep Red (Invitrogen, Cat.# T34077) experiment to measure the speed, length, and trajectory of EB1 comets.

### Patient-derived xenograft generation and treatment
Subcutaneous castration-resistant LuCaP35CR and LuCaP70CR PDX tumors were from Dr. Eva Corey (University of Washington, Seattle, WA)[39]. They were propagated in castrated male mice (6-8 week old ICRSC-M, IcrTac:ICR-Prkdcscid mice from Taconic) to generate an in vivo model of docetaxel resistance. Tumor size was assessed by caliper measurement. Treatment was initiated when the tumor volume reached 500 mm³. Mice were treated with 30 mg/kg docetaxel every 3 weeks by i.p. injection until they reached ~1500 – 2000 mm³. One last dose was then administered, and tumors were harvested two weeks after this last dose. For all experiments the light cycle was a 14-h light/10-h dark cycle. Temperature was 65–75 °F (~18–23 °C) with 40–60% humidity.

### LNCaP xenograft generation and treatment
Male nude mice (Taconic, NCRNU, age ~6 weeks) were injected subcutaneously with ~2 ×10⁶ cells from LNCaP cell lines stably overexpressing FOXJ1 or EV, which were suspended in PBS containing 50% Matrigel (Corning Basement Membrane Matrix, CLS 354234). Tumor size was assessed by caliper measurements. Treatment was initiated when tumors reached 500 mm³. Mice were treated with vehicle or 30 mg/kg docetaxel, i.p., on Day 9 and Day 21. Three tumors from the EV and FOXJ1_OE groups were harvested 3 days after the first treatment (Day 9). The remaining tumors were harvested at 3 days after the second treatment (Day 21). A portion of the tumors was frozen and embedded in Tissue-Tek O.C.T. compound for RNA sequencing, while the rest of the tumors were fixed in formalin and preserved in paraffin for immunofluorescence. A similar protocol was used to assess xenografts generated from LNCaP cells overexpressing TPPP3.

### RNA sequencing/low pass whole genome sequencing
Tumor tissue frozen blocks were sliced into 30 μm thick sections, totaling seven sections, and the first and seventh sections were stained with H&E. Once tumor quality was confirmed through H&E staining, the tissue sections were processed to extract RNA and DNA using the Allprep DNA/RNA mini kit (Qiagen, Q33238). RNA samples underwent RNA sequencing at GENEWIZ or Novogene, ensuring a minimum of 20 million read pairs per sample. DNA samples were sent to BGI America for low-pass whole-genomic sequencing.

### Western blot
Cells were lysed on ice using RIPA buffer containing phosphate and proteasome inhibitors (Roche). Samples were run on SDS-PAGE gel 4–15% (BioRad), and transferred to nitrocellulose membrane using Bio-Rad Trans-Blot Turbo Transfer System. Primary antibodies used include anti-GAPDH (Cell Signaling Technology, #2118, 1:3000), anti-Vinculin (Santa Cruz, sc–73614, 1:5,000), anti-Acetyl-α-Tubulin (Cell Signaling Technology, #5335, 1:3000), anti-α-Tubulin (Abcam, ab7291, 1:4000), anti-FOXJ1 (R&D Systems, AF3619, 1:100), anti-TPPP3 (Thermo Fisher, PA5–24925, 1:100), anti-phospho-Histone3 (ser10) (Cell Signaling Technology, #9701, 1:1000), anti-CLIP-170 (clone: E–8, Santa Cruz, 1:500). Secondary antibodies used include Anti-Rabbit-HRP conjugated antibody (Promega, W401B, 1:5000), Anti-Mouse- HRP conjugated antibody (Promega, W402B, 1:5000), and Anti-Goat-HRPconjugated antibody (R&D systems, A15999, 1:2000).

### qPCR
RNA was extracted using RNeasy mini kit (Qiagen). The qPCR reactions were prepared using TaqMan™ RNA- to-CT 1-Step Kit and run on ABI's step one plus 96-well real-time qPCR system. Taqman probes were ordered from Thermo Fisher including *FOXJ1* (Hs00230964_m1), *TPPP3* (Hs00372228_g1), *GMNC* (Hs04970928_m1), *HES1*(Hs00172878_m1), *HEY1* (Hs05047713_s1), and *GAPDH* control (4326317E).

### Tubulin polymerization assay
Tubulin polymerization assay was performed as described[22,40]. Briefly, cells were plated in 6-well plates and then treated with different concentrations of either docetaxel or cabazitaxel overnight. After the treatment, cells were washed with pre-warmed PBS and lysed at 37 °C for 10 min with 0.1 ml Low Salt buffer (20 mM Tris-HCl pH 6.8, 1 mM MgCl₂, 2 mM EGTA, 0.5% NP-40) supplemented with protease inhibitor cocktail (Roche). The lysates were centrifuged at 15,000 r.p.m. for 5 min at room temperature. The supernatants containing soluble tubulin (*S*) were transferred to another tube, separating them from the pellets containing polymerized tubulin (*P*). The pellets were resuspended into Low Salt Buffer and sonicated. Both supernatants and sonicated pellets were subjected to a Western blot.

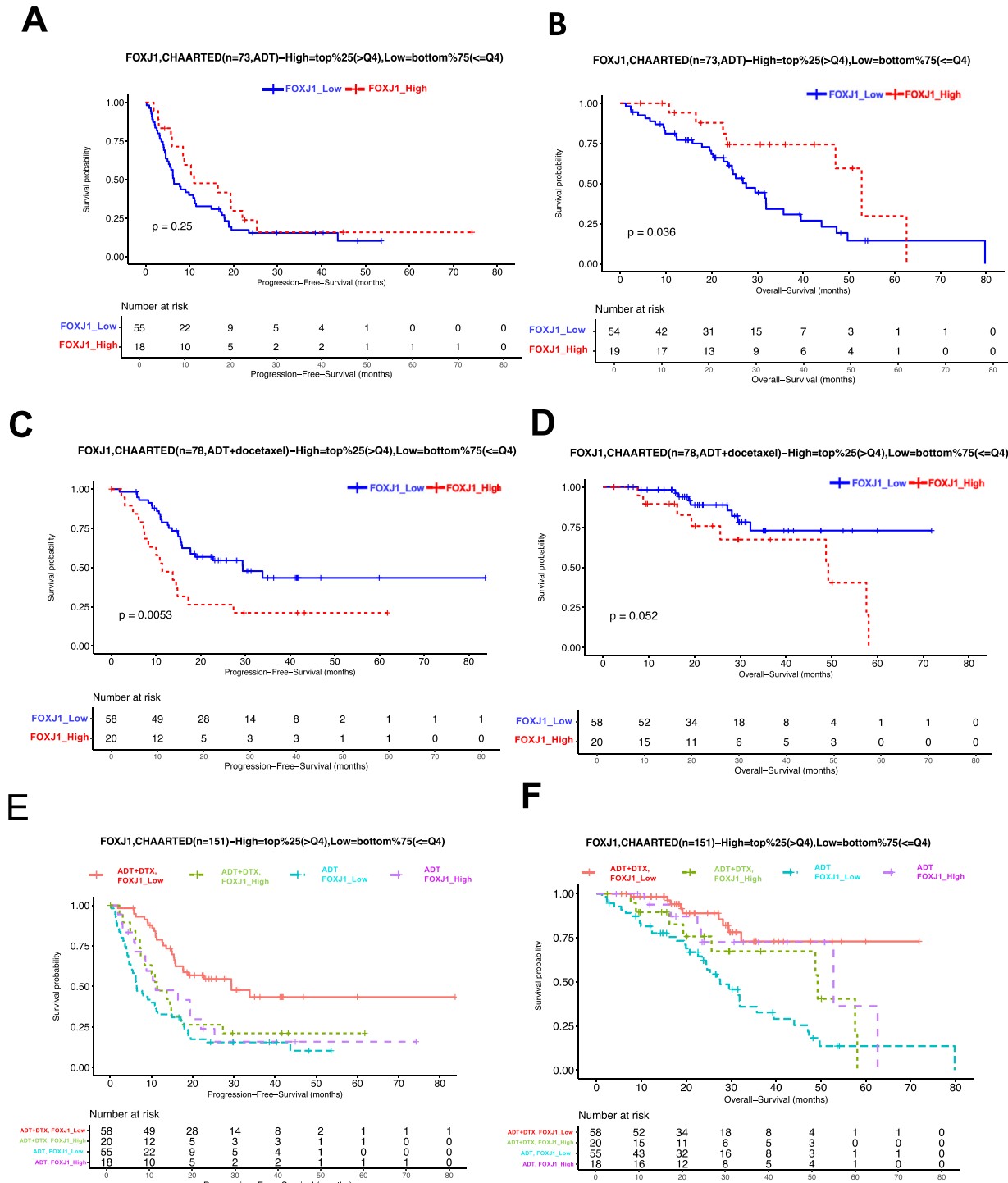

**Fig. 7 | Higher FOXJ1 in patient tumors is predictive of DTX resistance. A** Kaplan-Meier curves for progression-free survival of CHAARTED patients treated with ADT alone, stratified by FOXJ1 high expression (top 25%) versus low expression (bottom 75%). Significance for all comparisons was assessed by log-rank test. **B** Kaplan-Meier curves for overall survival of CHAARTED patients treated with ADT alone, stratified by FOXJ1 high expression (top 25%) versus low expression (bottom 75%). **C** Kaplan-Meier curves for progression-free survival of CHAARTED patients treated with ADT plus DTX, stratified by FOXJ1 high expression (top 25%) versus low expression

(bottom 75%). **D** Kaplan-Meier curves for overall survival of CHARRTED patients treated with ADT plus DTX, stratified by FOXJ1 high expression (top 25%) versus low expression (bottom 75%). **E** Kaplan-Meier curves comparing progression-free survival of CHAARTED patients treated with ADT alone or ADT plus DTX, stratified by FOXJ1 high expression (top 25%) versus low expression (bottom 75%). **F** Kaplan-Meier curves comparing overall survival of CHAARTED patients treated with ADT alone or ADT plus DTX with FOXJ1 high expression (top 25%) versus low expression (bottom 75%).

## Flow cytometry

PC cells were collected and fixed with ethanol at 4 °C, followed by PBS washing and Propidium Iodide (BioLegend) staining. Data were acquired with CytoFLEX LX (Beckman), and results were analyzed with FlowJo.

## Colony formation

A total of 5 × 105 PC cells were allocated into each well of 6-well plates and cultured overnight to allow attachment. The cells were then treated with DMSO vehicle or the indicated drugs for 7 days, after which the drugs were washed out and the cells were cultured in regular media for another 1-3 weeks until colonies were visible. The colonies were washed with PBS, fixed by pre-chilled methanol at −20 °C for 10 minutes, and followed by staining with crystal violet (Sigma Aldrich) solution (0.5% in 25% methanol) for 4 h at room temperature. After removing the crystal violet solution, the colonies were washed with distilled water 10 times until the background became clear. The colonies containing over 50 cells were counted. Individual colonies were quantified using Image J. For wells with confluent colonies, colony area and intensity were measured instead, using the Image J plugin *ColonyArea*[41].

## Immunofluorescence

Formalin-fixed, paraffin-embedded (FFPE) xenograft tumor sections were heated at 60 °C for 30 min prior to staining. Sections were deparaffinized in xylene, rehydrated through a graded ethanol series, and rinsed in phosphate-buffered saline (PBS). Heat-induced antigen retrieval was performed for 30 min using EnVision FLEX Target Retrieval Solution (High pH; Dako, DM828), followed by permeabilization with 0.3% Triton X-100 in PBS. Non-specific binding was blocked by incubation in 10% goat serum prepared in Tris-buffered saline (TBS).

Microtubules were immunolabeled using a rat monoclonal anti-β-tubulin antibody (clone YL1/2; EMD Millipore, Mab1864-I) diluted 1:100 in blocking buffer. After primary antibody incubation, sections were washed in TBS and incubated with Alexa Fluor 488–conjugated goat anti-rat secondary antibody (Thermo Fisher Scientific, A-11006) at a dilution of 1:500 for 1 h at room temperature. Slides were subsequently washed in TBS and counterstained with DAPI. Confocal images were acquired using a Zeiss LSM 700 microscope equipped with a ×63/1.4 NA oil-immersion objective (Zeiss, Germany). For assessment of taxane–microtubule interactions, Flutax residence time assays were performed on native cytoskeletons[34]. Cells were incubated with 1 μM FITC-conjugated paclitaxel (Flutax-2; Thermo Fisher Scientific, P22310) for 5 min at 37 °C, followed by rapid drug washout. Samples were immediately rinsed with PEMP buffer, and time-lapse imaging was conducted using a Zeiss spinning disk confocal microscope and a Nikon super-resolution microscope.

## Native cytoskeleton imaging

DU145 shNeg and DU145 FOXJ1 KD (shRNA_01) PC cells were permeabilised with 0.5% Triton X-100 for 60seconds, washed, and then treated with FITC-conjugated paclitaxel (Flutax-2) at a concentration of 1μM for 5minutes to label the microtubules, as previously described[42]. After treatment, the cells were washed to remove excess Flutax-2, and imaging was performed over a total duration of 90 minutes using a Zeiss spinning disk confocal microscope. Representative images of Flutax-2-labeled microtubules in native cytoskeletons from these cells were taken at timepoints 0, 15, 30, 60, and 90 minutes after washout. The fluorescence intensity of tubulin tracker was quantified using Fiji software[43].

## Immunofluorescence staining and imaging for cilia

BEAS-2B cells, a human bronchial epithelial cell line, were used as a positive control for cilia staining and were processed in parallel with LNCaP cells stably overexpressing FOXJ1 (LNCaP FOXJ1 OE).

Cells were initially plated in complete medium containing 10% fetal bovine serum (FBS). After allowing the cells to attach overnight, the medium was replaced with serum-free medium to induce serum starvation, which was maintained for an additional three days prior to fixation. Following serum starvation, cells were fixed and stained for acetylated α-tubulin, a well-established marker of stabilized microtubules in motile cilia. In addition, cells were stained with an anti-centrin antibody, which labels centrioles and basal bodies and was used in this context to identify the ciliary base and assess centriole localization. Nuclear DNA was counterstained with DAPI. Immunofluorescence imaging was performed using a Nikon structured illumination super-resolution microscope.

## EB1 imaging and quantification

DU145 shNeg and DU145 FOXJ1 KD (shRNA_01) PC cells stably expressing EB1-EGFP cells were washed and then treated with Tubulin Tracker Deep Red (Invitrogen, Cat.# T34077) at a concentration of 1 μM for 5 minutes to label the microtubules. After treatment, the cells were washed to remove excess tubulin tracker, and movies were acquired every 0.5 sec using a 60X lens in a Nikon Eclipse Ti2-E Inverted Microscope Yokogawa Spinning Disk Confocal System CSU-W1 SoRa. The fluorescence intensity of tubulin tracker and EB1 comets was quantified using the Nikon Elements software.

The speed of EB1 comets was analyzed using Nikon Elements software. EB1 movies were captured at a rate of one frame every 0.5 seconds for a total duration of 10 seconds. To accurately assess EB1 dynamics, areas within the leading edge of the cell were selected for quantification. These regions exhibited a relatively low density of EB1-EGFP staining and low background fluorescence. After selecting an area in the leading edge, the denoise function of the software was applied to reduce noise. Following this, the rolling ball algorithm was utilized to further enhance image quality. The defined threshold function was then set up to ensure consistent detection of EB1 comets. Using the bright spot function of the software, EB1 comets were identified and marked. The following parameters were measured: speed, number of comets, and EB1 trajectory. These measurements were conducted on hundreds of cells for each experimental condition.

The length of EB1-EGFP comets was measured in the leading edge of the cells, with a single time frame selected for this measurement. The same protocol was followed as previously described: the denoise function of the software was applied to reduce noise, followed by the application of the sharpen and rolling ball algorithms to enhance image quality. Subsequently, the threshold was defined to select all visible EB1 comets in the field. The border segmentation function was used to separate individual comets. The length of each EB1 comet was then computed and quantified using the software. To generate a bar graph representing the number of comets for each EB1 comet length, the automated measurement results function was utilized.

## Image analysis and quantitative assessment of effective drug-target engagement

We previously developed a quantitative measure of effective drug-target engagement (DTE) as a marker of taxane response or resistance, by developing a scoring algorithm to quantify MT fluorescent intensity following indirect immunofluorescence and confocal microscopy imaging[30]. Briefly, tubulin immunofluorescence microscopy in cells or tissues was performed under identical conditions and the extent of a tubulin fluorescence intensity was quantified by the number of pixels in the top quartile of integrated intensity and normalized by the total cell count, as determined by the DAPI nuclear stain. This scoring algorithm was validated in biopsies from patients with PC following taxane treatment, where we found that high DTE correlated with taxane sensitivity and vice versa[30].

## Statistics and reproducibility

All the experiments were randomised, and no data were excluded from the analyses. Mice were randomly allocated into treatment groups, and tumor proliferation were displaced as curves with each point representing mean ± SEM. Two-way ANOVA tests with were applied to the animal experiments. Student's t-test were applied to colony formation, cell cycle experiments, and tubulin polymerization assay. Microtubule fluorescence intensity (MFI) quantitation, as a readout of DTE, is displayed as a dot plot with mean ± SEM values. Mann–Whitney tests were applied to the details of each analysis, and significance thresholds are listed in each figure. Nuclear AR single-cell intensity measurements were analyzed using one-way ANOVA with Dunnett's post hoc test for within-cell-line comparisons and two-tailed independent-samples t-tests for comparisons between LNCaP EV and LNCaP FOXJ1 OE cells. GraphPad Prism 10 was used for the analyses. Overall significance threshold is $p < 0.05$. Gene expression analysis of TCGA primary PC samples (Firehose Legacy) were analyzed on cBioPortal[44]. Whole genome sequencing data were visualized using the Integrative Genomics Viewer[45]. Correlations between FOXJ1 and docetaxel sensitivity across cell lines was determined using the Cancer Dependency Map (DepMap)[46].

## Reporting summary

Further information on research design is available in the Nature Portfolio Reporting Summary linked to this article.

## Data availability

Source data are provided with this paper. RNA-seq data generated in this study have been deposited in GEO under accession number GSE276579. Low pass whole genome sequencing data generated in this study have been deposited in the NCI Sequence Read Archive under accession number PRJNA1157875 (https://www.ncbi.nlm.nih.gov/sra). Source data for plots are provided with this paper in an Excel file and uncropped blots are included in the Supplementary Figs. Source data are provided with this paper.

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

## Acknowledgements

This work was supported by grants from the NIH, including R01 CA266704 (SPB, PG), P50 CA272390 (SPB, LP, DJE), P01 CA163227 (SPB, EC), R01CA228512 (PG), P50 CA97186 (EC), from DoD W81XWH-19-1-0666 (PG), and from a BIDMC GU research pilot award (FX). LP and DJE had support from Prostate Cancer Foundation (PCF) Young Investigator Awards, and SPB from a PCF Challenge Award. AV had support from DoD (Physician Research Award, PC200820) and ASCO (Young Investigator Award, 2021A010981).

## Author contributions

Conceptualization: R.S.B., P.G., S.P.B.; Data curation: F.X., A.G., P.G., and S.P.B.; Formal Analysis: F.X., A.G., D.F., M.L., L.P., D.J.E., P.G., and S.P.B.; Funding acquisition: P.G. and S.P.B.; Investigation: F.X., A.G., B.E., C.M.D., O.V., A.Ga.d., A.V.; Methodology: F.X., A.G., P.G.; Project administration: P.G., S.P.B.; Resources: E.C., P.G., S.P.B.; Supervision: P.G., S.P.B.; Validation: F.X., A.G., P.G., SPB Visualization: F.X., A.G., P.G., S.P.B.; Writing original draft: S.P.B.; Writing edits: F.X., A.G., P.G.

## Competing interests

The authors declare no competing interests.
