## [Transparent Peer Review file · Nature Communications]

FOXJ1 mediates taxane resistance through regulation of microtubule dynamics

Corresponding Author: Dr Steven Balk

Version 0:

Reviewer comments:

Reviewer #1

(Remarks to the Author)

This is an innovative manuscript coming from Dr. Balk's group, focusing on the mechanistic dissection of therapeutic resistance to microtubule-targeting (taxane) chemotherapy in in-vivo relevant models of prostate cancer. The authors report novel results establishing the mechanistic link between the FOXJ1 signaling axis, microtubule dynamics and taxane resistance. The focus on FOXJ1 as the protagonist takes significant translational dimensions as clinically FOXJ1 gene amplification was elevated in taxane-treated prostate cancer patients. Moreover, in the CHAARTED clinical trial of docetaxel combined with (androgen deprivation) ADT for metastatic prostate cancer higher baseline FOXJ1 was predictive of decreased survival in patients treated with docetaxel (1st line taxane chemotherapy). The findings validated in pre-clinical model of advanced prostate cancer identify a new clinically impactful mechanism of taxane resistance, exploitation of which will lead to a potential stratification of patients likely to develop resistance to taxane treatment, thus should be considered for alternative therapeutic modalities.

The study is well-designed built on a compelling rationale and the results significantly impactful at the mechanistic, translational and clinical level. The work is beautifully executed supporting new evidence on mechanisms of resistance to taxane chemotherapy in advanced prostate cancer.

I only have a few minor points to improve presentation of the manuscript, as follows:

- 1) The length of the manuscript can be shortened, in order to allow for a more succinct flow. Experimental details of standard methods can be deleted for clarity and space.
- 2) A schematic diagram indicating the FOXJ1 signaling interactions with other major effectors of the microtubule dynamics and the androgen receptor (AR) signaling mechanism, will benefit the interpretation of the results by the reader. It will be a significant addition.
- 3) Can the authors comment on the role/contribution of FOXJ1 to therapeutic resistance in response to other chemotherapeutic strategies (e.g. PARP inhibitorS?_

Reviewer #2

(Remarks to the Author)

Key results:

Biomarkers to predict docetaxel sensitivity for men with prostate cancer are lacking. The authors of this manuscript use multiple methodologies to demonstrate the FoxJ1 expression increases intrinsic resistance to docetaxel through modulation of MT function. In general, this is a well-developed manuscript using sound experimental approaches of potential great importance to clinicians treating men with metastatic prostate cancer.

Validity:

The data and conclusions are robust and valid. I would like to see more integration of FoxJ1 expression status with the currently available prognostic and predictive biomarkers Rb, PTEN, and p53 based on the CHAARTED data.

Significance:

FoxJ1 expression may eventually become an important predictive biomarker for taxane benefit in prostate and potentially fulfills an unmet need for men with metastatic prostate cancer.

Data and methodology:

All figures were reviewed including supplementary information were reviewed and deemed appropriate.

The authors begin by developing two castration-resistant PDXs. In one of these, LuCaP35CR PDX, GEMC was the most significantly enriched gene. Given that downstream FOXJ1 controls expression of MT associated protein and is enriched in GOPB gene sets, it was further evaluated. TCGA Mining was performed and consistent with the PDX. DepMap drug response data indicated that FOXJ1 expression was positively correlated with docetaxel resistance in cancer cell lines and gene amplification was increased two-fold in taxane-treated patients.

Next, docetaxel treated LNCaP cells stably overexpressing FOXJ1 vs vector were subjected to colony formation (showing less growth) and cell sorting (less G2/M indicative of less docetaxel induced mitotic arrest). pH3S10 was not increased also indicating less mitosis in docetaxel treated FOXJ1 overexpressing cells. MT aggregation was impaired in docetaxel treated LNCaP cells stably overexpressing FOXJ1 but basal levels were increased.

LNCaP, DU145 FOXJ1 KD cells demonstrated increased docetaxel sensitivity and induced MT bundling as well as more extensive labeling of MT network, measured by BFI. To understand the mechanism of bundling, docetaxel induced alpha-tubulin acetylation in KD vs parental cells was examined and found to be dramatically enhanced (8.7 x vs 1.8 x) upon docetaxel treatment. MT dynamics were evaluated by EB1 comet assay in FOXJ1 KD vs control cells with comets demonstrating more pronounced taxane-mediated decrease in KD cells. EB1 length but not number was decreased indicative of slower MT growth rate in KD cells and confirmed cinematographically for resting and taxane treated cells.

LNCaP xenografts overexpressing FOXJ1 cells exhibited docetaxel resistance and decreased mitotic figures.

Downstream Pathways were assessed with mitotic spindle being most enriched gene set and findings were congruent with LuCaP35CR. TPPP3 overexpressing LNCaP cells were generated with findings similar to those of FOXJ1.

Time courses of docetaxel treatment in cultured cells indicated induction of FOXJ1 and TPPP3 from 24-96 hours with authors showing it independent of cell cycle dependent upregulation. FOXJ1 induction was shown to be preceded 12 hours by upstream GEMC expression upon docetaxel treatment

Finally, using CHARTED data, quartile analysis of FOXJ1 expression was shown to inversely predict addition of docetaxel benefit with benefit seen more pronounced in ADT alone arm. It was not overall prognostic except possibly in high grade disease.

Suggested improvements:

Line 86: Docetaxel is not the only chemotherapy used in PC treatment. Would say it is primary first line chemotherapy.

The ABCB1/MDR PDX results were identified but not at all discussed- are the ABC1/MDR overexpression and GEMC/FOXJ1 pathways mutually exclusive resistance pathways? Which is predominant in patients. Is ABCB1 at all altered by FOXJ1 expression.

Would like more discussion of potential mechanisms downstream of GEMC/FOXJ1/TPPP3 axis- How exactly does this lead to resistance. Is there a connection with AR translocation or classic downstream targets such as bcl-2, or is it an off-target effect. Experimentally, would make your study more compelling to include the above targets.

Would like a discussion how FOXJ1 compares to other established prognostic alterations such as PTEN, Rb and p53, which are increasingly being used by clinicians to select patients for taxane-based therapy. Was there any evidence of any correlation of FOXJ1 expression with these more established markers?

Is there any correlation between timing (synchronous vs metachronous) and volume in the CHARTED study as these are both integral to NCCN guidelines for the use of docetaxel in the CSPC setting.

Does ABCB1 expression in the CHARTED study have any predictive value- suggest including in discussion.

References:

Adequate and appropriate.

Reviewer #3

(Remarks to the Author)

In the paper by Balk and colleagues, data are presented that suggest an induction of the motile ciliated transcriptional program, notably that driven by the forkhead transcription factor FOXJ1 in docetaxel resistant prostrate cancer (PC) cells. The authors have also presented results that indicate FOXJ1 mediated alterations in microtubule dynamics as a likely cause of docetaxel resistance and implicate the FOXJ1 target gene and tubulin associated protein, tubulin polymerization promoting protein 3 (TPPP3) as one of the possible downstream effectors of tubulin dynamics that mediates docetaxel resistance.

Ectopic activation of the motile, and more specifically, the multiciliated cell transcriptional program in docetaxel-resistant PC is an intriguing observation and this pathway provides a mechanistic basis of how PC can become resistant to taxane treatment.

Criticisms:

a) The authors state that not much is known about the function of FOXJ1 in non-ciliated cells. This is an erroneous belief. FOXJ1 is largely a motile ciliated cell-specific transcription factor. It is expressed not only in motile multiciliated cells as the authors repeatedly reference, but it be noted that FOXJ1 is also expressed in cells that differentiate monomotile cilia such as in the vertebrate left-right organizer and spermatozoa. Besides these motile cilia bearing cell-types, there are very few other cells where FOXJ1 is expressed in vivo. So, the notion that not much is known about this protein in non-ciliated cells is wrong.

b) It is not clear how the multiciliated cell transcriptional program gets activated on docetaxel administration. The authors invoke attenuation of Notch signaling but the data presented are rather superficial. In normal development, Notch signaling

acts cell non-autonomously to inhibit the multiciliated cell program. It is not clear in what capacity the authors envisage docetaxel could be influencing Notch signaling. In my view, this is a key area that needs more detailed investigation.

c) Even though key multiciliated cell transcription factors like GMNC and FOXJ1 are over-expressed, the data do not indicate any sign of cilia formation or multiciliation. Have the authors looked at these issues carefully? Are centrioles amplified these cells as it happens in multiciliated cells undergoing differentiation? Do they see any kind of ciliation – mono or multiple?

Version 1:

Reviewer comments:

Reviewer #1

(Remarks to the Author)

This is a novel revised manuscript, focusing on novel mechanisms driving therapeutic resistance to microtubule-targeting (taxane) chemotherapy in models of advanced prostate cancer. The authors provide an insightful response to the previous critique and new compelling data on mechanistic contribution of FOXJ1 signaling axis towards impacting microtubule dynamics and taxane resistance. The findings are of major translational significance in defining novel biomarkers for resistance to taxane chemotherapy. The work is of high impact and relevance to the prostate cancer investigators.

Reviewer #3

(Remarks to the Author)

The authors have addressed my criticisms to the best of their abilities.

Reviewer #4

(Remarks to the Author)

RESPONSES TO REVIEWER COMMENTS

Reviewer #1 (Remarks to the Author):

This is an innovative manuscript coming from Dr. Balk's group, focusing on the mechanistic dissection of therapeutic resistance to microtubule-targeting (taxane) chemotherapy in in-vivo relevant models of prostate cancer. The authors report novel results establishing the mechanistic link between the FOXJ1 signaling axis, microtubule dynamics and taxane resistance. The focus on FOXJ1 as the protagonist takes significant translational dimensions as clinically FOXJ1 gene amplification was elevated in taxane-treated prostate cancer patients. Moreover, in the CHARTED clinical trial of docetaxel combined with (androgen deprivation) ADT for metastatic prostate cancer higher baseline FOXJ1 was predictive of decreased survival in patients treated with docetaxel (1st line taxane chemotherapy). The findings validated in pre-clinical model of advanced prostate cancer identify a new clinically impactful mechanism of taxane resistance, exploitation of which will lead to a potential stratification of patients likely to develop resistance to taxane treatment, thus should be considered for alternative therapeutic modalities.

The study is well-designed built on a compelling rationale and the results significantly impactful at the mechanistic, translational and clinical level. The work is beautifully executed supporting new evidence on mechanisms of resistance to taxane chemotherapy in advanced prostate cancer. I only have a few minor points to improve presentation of the manuscript, as follows:

Response: We thank the reviewer for the positive feedback and the constructive suggestions, which we address below.

1) The length of the manuscript can be shortened, in order to allow for a more succinct flow. Experimental details of standard methods can be deleted for clarity and space.

Response: Thank you for the suggestion. We have done editing to remove standard methods and make the manuscript more succinct.

2) A schematic diagram indicating the FOXJ1 signaling interactions with other major effectors of the microtubule dynamics and the androgen receptor (AR) signaling mechanism, will benefit the interpretation of the results by the reader. It will be a significant addition.

Response: Thank you also for this suggestion. We have added a diagram showing FOXJ1 acting as a transcription factor to increase expression of TPP3 and additional microtubule associated proteins, with a subsequent increase in basal microtubule dynamics (including increased AR nuclear import) and decreased taxane binding (Figures S26).

Figure S26. Summary of FOXJ1 effects on MT dynamics and response to taxanes.

3) Can the authors comment on the role/contribution of FOXJ1 to therapeutic resistance in response to other chemotherapeutic strategies (e.g. PARP inhibitors?)

Response: We had shown that increased FOXJ1 decreases the responses to cabazitaxel as well as to docetaxel. However, it is an excellent suggestion to examine other agents including those that interact with microtubules or act by other mechanisms. Hence, we have carried out colony formation assays to assess responses to a series of agents that target microtubules by distinct mechanisms (colchicine, vincristine, epothilone, and maytansine). We also examined agents that act by other mechanisms (olaparib, cisplatin, and carboplatin). As shown in the figure (new Figure S6), FOXJ1 overexpression conferred resistance to each of the microtubule targeted agents. Notably, the binding sites for these

are distinct from the taxane site, suggesting that the alteration in microtubules in response to FOXJ1 overexpression does not specifically affect the taxane binding site. Interestingly, cells overexpressing FOXJ1 also had decreased sensitivity to cisplatin, carboplatin, and olaparib, although the effect was less substantial than for the MT targeted agents. Further studies are needed to determine whether the basis for this is related to increases in microtubule dynamics (possibly through increased drug efflux or other mechanisms), or is through distinct mechanisms.

Reviewer #2 (Remarks to the Author):

Key results:

Biomarkers to predict docetaxel sensitivity for men with prostate cancer are lacking. The authors of this manuscript use multiple methodologies to demonstrate the FoxJ1 expression increases intrinsic resistance to docetaxel through modulation of MT function. In general, this is a well-developed manuscript using sound experimental approaches of potential great importance to clinicians treating men with metastatic prostate cancer.

Response: We thank the reviewer for their overall positive evaluation of the work.

Figure S6. FOXJ1 overexpression decreases sensitivity to additional tubulin target drugs and additional agents. (A) LNCaP cells overexpressing FOXJ1 or empty vector (EV) control were plated at high density and treated for 7 days with the indicated drugs. The drugs were then washed out and cells were cultured for an additional 1-3 weeks until colonies were visible. Colonies were then stained with crystal violet and quantified. (B) Cells as in (A) were treated as indicated.

Validity:

The data and conclusions are robust and valid. I would like to see more integration of FoxJ1 expression status with the currently available prognostic and predictive biomarkers Rb, PTEN, and p53 based on the CHARTED data.

Response: This is an excellent suggestion. One way we have addressed it is to determine whether there is a correlation between expression of FOXJ1 and RB1, PTEN, or TP53. As shown in the figure (new Figure S20), analysis of the TCGA primary PCa data (which reflects the population treated in the CHARTED trial) does show a weak negative correlation between expression of FOXJ1 and each of these tumor suppressor genes. While it is possible that they are directly or indirectly regulated by FOXJ1, it is more likely that decreased expression of each of these genes can increase FOXJ1 indirectly through effects on cell cycle. In any case, the presence of genomic alterations in these tumor suppressor genes are more prognostic and predictive than levels of expression. Therefore, the figure also shows mutations (left panels) and copy number alterations in PTEN, TP53, and RB1 (right panels). This analysis does not reveal an association between genomic alterations in these genes and increased FOXJ1. Overall these findings do not indicate that increased FOXJ1 expression is driven by oncogenic alterations in these tumor suppressor genes.

Figure S20. Correlations between FOXJ1 and PTEN, TP53, and RB1 in primary PC. Data on left panels shows mRNA levels and mutation status for PTEN, TP53, and RB1. Data on right panels show copy number alterations. Data are from the TCGA Firehose Legacy and were analyzed on cBioPortal.

A second approach we have taken is to analyze the gene expression data from CHARTED to determine whether expression of RB1, PTEN, or TP53 are associated with response to docetaxel. As shown in the figure (new Figure S25), in contrast to the results for FOXJ1, expression of these genes was not associated with survival. However, one caveat is that mutations or indels may also result in decreased activity for these tumor suppressor genes, which may not be reflected in decreased transcript levels. Therefore, we cannot rule out entirely that decreased activity of these genes has effects on the response to docetaxel (although the data above indicate that any such

effect would be independent of altered FOXJ1 expression). We also examined ABCB1 expression, and did not find a negative correlation between increased expression and response. Notably, this does not argue against increased ABCB1 expression as a basis for acquired resistance, as the gene expression analyses were done on tumor samples taken prior to initiation of therapy.

Figure S25. Kaplan-Meier curves illustrating OS in CHARTED patients based on expression of RB1, TP53, PTEN, and ABCB1.

Significance:

FoxJ1 expression may eventually become an important predictive biomarker for taxane benefit in prostate and potentially fulfills an unmet need for men with metastatic prostate cancer.

Response: We appreciate these comments.

Data and methodology:

All figures were reviewed including supplementary information were reviewed and deemed appropriate. The authors begin by developing two castration-resistant PDXs. In one of these, LuCaP35CR PDX, GEMC was the most significantly enriched gene. Given that downstream FOXJ1 controls expression of MT associated protein and is enriched in GOPB gene sets, it was further evaluated. TCGA Mining was performed and consistent with the PDX. DepMap drug response data indicated that FOXJ1 expression was positively correlated with docetaxel resistance in cancer cell lines and gene amplification was increased two-fold in taxane-treated patients.

Next, docetaxel treated LNCaP cells stably overexpressing FOXJ1 vs vector were subjected to colony formation (showing less growth) and cell sorting (less G2/M indicative of less docetaxel induced mitotic arrest). pH3S10 was not increased also indicating less mitosis in docetaxel treated FOXJ1 overexpressing cells. MT aggregation was impaired in docetaxel treated LNCaP cells stably overexpressing FOXJ1 but basal levels were increased.

LNCaP, DU145 FOXJ1 KD cells demonstrated increased docetaxel sensitivity and induced MT bundling as well as more extensive labeling of MT network, measured by BFI. To understand the

mechanism of bundling, docetaxel induced alpha-tubulin acetylation in KD vs parental cells was examined and found to be dramatically enhanced (8.7 x vs 1.8 x) upon docetaxel treatment. MT dynamics were evaluated by EB1 comet assay in FOXJ1 KD vs control cells with comets demonstrating more pronounced taxane-mediated decrease in KD cells. EB1 length but not number was decreased indicative of slower MT growth rate in KD cells and confirmed cinematographically for resting and taxane treated cells. LNCaP xenografts overexpressing FOXJ1 cells exhibited docetaxel resistance and decreased mitotic figures. Downstream Pathways were assessed with mitotic spindle being most enriched gene set and findings were congruent with LuCaP35CR. TPPP3 overexpressing LNCaP cells were generated with findings similar to those of FOXJ1.

Time courses of docetaxel treatment in cultured cells indicated induction of FOXJ1 and TPPP3 from 24-96 hours with authors showing it independent of cell cycle dependent upregulation. FOXJ1 induction was shown to be preceded 12 hours by upstream GEMC expression upon docetaxel treatment. Finally, using CHARTED data, quartile analysis of FOXJ1 expression was shown to inversely predict addition of docetaxel benefit with benefit seen more pronounced in ADT alone arm. It was not overall prognostic except possibly in high grade disease.

Suggested improvements:

Line 86: Docetaxel is not the only chemotherapy used in PC treatment. Would say it is primary first line chemotherapy.

Response: Thank you for pointing this out. We did indeed mean to say it is first line. This has been changed.

The ABCB1/MDR PDX results were identified but not at all discussed- are the ABCB1/MDR overexpression and GEMC/FOXJ1 pathways mutually exclusive resistance pathways? Which is predominant in patients. Is ABCB1 at all altered by FOXJ1 expression.

Response: The reviewer is correct that while we found increased ABCB1 in one docetaxel resistant model, we did not pursue this mechanism or discuss it further (except to say that its clinical significance remains to be established) as it has been extensively studied and we wanted to focus on the novel FOXJ1 results. Whether the FOXJ1 and ABCB1 pathways are mutually exclusive is an interesting question. There is no clear mechanistic link that would suggest mutual exclusivity, although it is not unlikely that cells with ABCB1 amplification would no longer be under strong pressure to select for other alterations that cause taxane resistance. Similarly, cells with FOXJ1 upregulation or amplification may be under less selective pressure to increase ABCB1. Notably, analysis of the SU2C Dream Team data in cBioPortal (see below) shows that some metastatic CRPC cases have amplification of both FOXJ1 and ABCB1.

However, only two of the ABCB1 amplified tumors were taxane exposed, with one having low FOXJ1 and one having high FOXJ1 (see below), so it is difficult to draw any definite conclusions about mutual exclusivity.

As noted above, the extent to which ABCB1 is a driver of taxane resistance in patients is not clear, so it is difficult to assess whether it or FOXJ1 is more prominent. However, while FOXJ1 amplification is more frequent in taxane exposed versus naïve CRPC cases (see Figure 1H), ABCB1 amplification is comparable in both (4.36 in naïve and 5.44% in exposed, see below). This has now been added as a new figure (**Figure S1B**).

Finally, we have no data indicating that ABCB1 expression is regulated by FOXJ1. We noted in the results that ABCB1 was not increased in the docetaxel resistant LuCaP35CR xenografts. Moreover, it was not increased in cells with ectopic FOXJ1 overexpression. Finally, as shown in the figure on the right, there is no correlation between increased FOXJ1 and ABCB1 in clinical samples.

Would like more discussion of potential mechanisms downstream of GEMC/FOXJ1/TPPP3 axis- How exactly does this lead to resistance. Is there a connection with AR translocation or classic

downstream targets such as bcl-2, or is it an off-target effect. Experimentally, would make your study more compelling to include the above targets.

Response: The major conclusion we reach in the manuscript is that an increase in the GEMC/FOXJ1/TPPP3 axis confers resistance by altering microtubule dynamics. This is supported by multiple lines of evidence including the direct demonstration that FOXJ1 overexpression decreases docetaxel-mediated microtubule stabilization. Moreover, we directly show FOXJ1 downregulation results in increased taxane binding to microtubules. However, as noted by the reviewer, we do not discuss precisely why mitigating taxane-mediated microtubule stabilization confers resistance (or conversely why taxane-mediated microtubule stabilization kills cells). Our presumption (which is supported by the literature) is that multiple pathways are impaired by disruption of microtubule dynamics. One is clearly mitosis due to effects on the mitotic spindle, and we show that FOXJ1 overexpression mitigates the taxane-mediated mitotic arrest.

However, as noted by the reviewer, increased FOXJ1 likely mitigates other toxic effects of taxanes. We believe it is beyond the scope of this study to comprehensively examine the many downstream effects of interfering with microtubule function. However, as suggested by the reviewer, we did examine effects of FOXJ1 overexpression on AR nuclear translocation. As show in a new figure (Figure S11), and consistent with previous reports from us and others, taxane treatment decreased the androgen-stimulated nuclear localization of AR. Notably, FOXJ1 overexpression significantly increased this androgen-stimulated nuclear localization, which is consistent with our finding that FOXJ1 overexpression increases microtubule dynamics. However, docetaxel still decreased AR nuclear localization in the FOXJ1 overexpressing cells. One interpretation of this result is that the microtubule-dependent transport of AR is mediated by specific cofactors that are particularly sensitive to the effects of taxanes, but further studies are clearly needed to test this or alternative hypotheses.

Figure S11. FOXJ1 overexpression increases AR nuclear localization in response to androgen. Quantification of nuclear AR levels from immunofluorescence images. Data are presented as mean \pm SD. One-Way ANOVA was used to compare AR levels within each cell line's drug treatment groups. Independent samples t-tests were performed for comparisons between the LNCaP parental and LNCaP FOXJ1 OE cell lines. In both LNCaP parental and LNCaP FOXJ1 OE cell lines, R1881 treatment significantly increased nuclear AR levels (due to nuclear accumulation) compared to untreated controls. Subsequent co-treatment with DTX (10 nM and 100 nM) significantly reduced the R1881-induced nuclear AR in both cell lines, with the 100 nM DTX dose showing a stronger suppressive effect. While no significant difference was observed in nuclear AR between untreated LNCaP parental and LNCaP FOXJ1 OE cell lines, LNCaP FOXJ1 OE cells treated with R1881 exhibited significantly higher nuclear AR levels compared to LNCaP parental cells treated with R1881.

Finally, as noted by the reviewer, AR targeted therapies may directly or indirectly increase BCL2. Therefore, it is possible that FOXJ1 could be acting in part indirectly through AR to support its nuclear translocation and mitigate decreased AR activity in response to taxane. However, this would presumably result in decreased BCL2. FOXJ1 could certainly also have effects on BCL2 or other apoptosis related genes independent of AR. However, the expression of BCL2 is not correlated with FOXJ1 expression in clinical data sets (see panel on right), and we did not find significantly increased expression of BCL2 or other BCL2 family anti-apoptotic proteins in FOXJ1 overexpressing cells. Notably, while taxanes can impair AR nuclear translocation, they clearly interfere with myriad other cellular functions, so effects on AR are probably not the major driver of their efficacy. Therefore, we have not carried out further AR focused studies.

Would like a discussion how FOXJ1 compares to other established prognostic alterations such as PTEN, Rb and p53, which are increasingly being used by clinicians to select patients for taxane-based therapy. Was there any evidence of any correlation of FOXJ1 expression with these more established markers?

Response: This comment is related to the suggestion above under the topic “Validity”. As described in that response, we did not find evidence that loss of these tumor suppressors was correlated with FOXJ1 expression. We also examined the CHAARTED data and did not find any correlations between response and expression of PTEN, RB1, or TP53 (see above).

Is there any correlation between timing (synchronous vs metachronous) and volume in the CHAARTED study as these are both integral to NCCN guidelines for the use of docetaxel in the CSPC setting.

Response: This is a great question. It is certainly possible that FOXJ1 expression in CHAARTED may be associated with tumor volume or other clinical features, and/or that integrating these clinical data with FOXJ1 expression would enhance predictive value. Unfortunately we don’t have this data for the CHAARTED study, and it would probably be underpowered to address this point. However, as described in the manuscript, FOXJ1 expression does not appear to be prognostic in primary PC or CRPC. To further address this question we also examined biochemical recurrence (BCR) in TCGA primary tumors as a metric that is presumably associated with occult metastatic disease. Notably, there was no correlation between FOXJ1 and BCR (see panel on right).

Does ABCB1 expression in the CHARTED study have any predictive value- suggest including in discussion.

Response: *This is a great question. As described in the response to the “Validity” section, we found that ABCB1 expression did not have predictive value in CHARTED. As noted above, this data is included in a new Figure (Figure S25) and in the Discussion.*

References:

Adequate and appropriate.

Reviewer #3 (Remarks to the Author):

In the paper by Balk and colleagues, data are presented that suggest an induction of the motile ciliated transcriptional program, notably that driven by the forkhead transcription factor FOXJ1 in docetaxel resistant prostrate cancer (PC) cells. The authors have also presented results that indicate FOXJ1 mediated alterations in microtubule dynamics as a likely cause of docetaxel resistance and implicate the FOXJ1 target gene and tubulin associated protein, tubulin polymerization promoting protein 3 (TPPP3) as one of the possible downstream effectors of tubulin dynamics that mediates docetaxel resistance.

Ectopic activation of the motile, and more specifically, the multiciliated cell transcriptional program in docetaxel-resistant PC is an intriguing observation and this pathway provides a mechanistic basis of how PC can become resistant to taxane treatment.

Criticisms:

a) The authors state that not much is known about the function of FOXJ1 in non-ciliated cells. This is an erroneous belief. FOXJ1 is largely a motile ciliated cell-specific transcription factor. It is expressed not only in motile multiciliated cells as the authors repeatedly reference, but it be noted that FOXJ1 is also expressed in cells that differentiate monomotile cilia such as in the vertebrate left-right organizer and spermatozoa. Besides these motile cilia bearing cell-types, there are very few other cells where FOXJ1 is expressed in vivo. So, the notion that not much is known about this protein in non-ciliated cells is wrong.

Response: *We thank the reviewer for pointing this out. We agree that it was misleading to indicate that not much is known about FOXJ1 in non-ciliated cells. While it has perhaps been most studied in the development of multi-ciliated cells, it is also expressed in cells that differentiate mono-motile cilia (as noted by the reviewer). Moreover, there have been many studies describing its expression in nonciliated cells including tumor cells, although its functions in these cells have not been well studied. We have now clarified these points and added a reference in the revised manuscript.*

b) It is not clear how the multiciliated cell transcriptional program gets activated on docetaxel administration. The authors invoke attenuation of Notch signaling but the data presented are rather superficial. In normal development, Notch signaling acts cell non-autonomously to inhibit the multiciliated cell program. It is not clear in what capacity the authors envisage docetaxel could be influencing Notch signaling. In my view, this is a key area that needs more detailed investigation.

Response: *We apologize for any confusion on this point. We investigated NOTCH because (as noted by the reviewer) a decrease in NOTCH signaling in the microenvironment initiates GEMC1 expression in precursor multiciliated cells. Moreover, we found in the TCGA PC gene data set*

that *GEMC1* expression was highly negatively correlated with expression of the Notch-induced genes *HES1* and *HEY1*, suggesting that docetaxel may be inducing *GEMC1* by decreasing Notch signaling (see previous Figure S13B). Therefore, we examined Notch signaling in response to docetaxel. However, we **did not** find a decrease in *HES1* or *HEY1* expression in response to docetaxel (see previous Figure S13C). Based on this we concluded that the initial increase in *GEMC1* **is not** mediated by a decrease in Notch. Therefore, we have not explored further whether, and by what mechanisms, docetaxel may influence Notch signaling. Nonetheless, we agree this area needs further investigation given the negative correlation between *GEMC1* and *HES1/HEY1*. Indeed, it is possible that while short term docetaxel treatment does not directly suppress Notch, its longer term effects *in vivo* may be to directly or indirectly suppress Notch and therapy further induce *GEMC1* and *FOXJ1*. We have now added these points to the Discussion.

c) Even though key multiciliated cell transcription factors like *GMNC* and *FOXJ1* are over-expressed, the data do not indicate any sign of cilia formation or multiciliation. Have the authors looked at these issues carefully? Are centrioles amplified these cells as it happens in multiciliated cells undergoing differentiation? Do they see any kind of ciliation – mono or multiple?

Response: This is also a great question. Indeed, during the early stages of this study we did look for evidence of cilia formation in cells overexpressing *FOXJ1*. We did not see any such evidence, but did not have positive controls in these studies. Therefore, in response to the reviewer's comment we have gone back and repeated these experiments with positive controls. As shown in the figure below, cilia are clearly seen by staining for acetylated tubulin in BEAS-2B cells, a human bronchial epithelial cell line. In contrast, we could not detect cilia in LNCaP cells overexpressing *FOXJ1* (which were process in parallel with the BEAS-2B cells). We also see no evidence of of centriole amplification. These results have been added to the revised manuscript (Figure S7).